

# Tsunami run-up estimation based on a hybrid numerical flume and a parameterization of real topobathymetric profiles

Íñigo Aniel-Quiroga[1], Omar Quetzalcóatl[1], Mauricio González[1], Louise Guillou[1]

[1]Environmental Hydraulics Institute, Universidad de Cantabria - Avda. Isabel Torres, 15, Parque Científico y Tecnológico de Cantabria, 39011, Santander, Spain

*Correspondence to*: Íñigo Aniel-Quiroga (anieli@unican.es)

**Abstract.** Tsunami run-up is a key value to determine when calculating and assessing the tsunami hazard in a tsunami-prone area. Run-up is accurately calculated by means of numerical models, but these models require high-resolution topobathymetric data, which are not always available, and long computational times. These drawbacks restrict the application of these models to the assessment of small areas. As an alternative method, to address large areas, empirical formulae are commonly applied to estimate run-up. These formulae are based on numerical or physical experiments on idealized geometries. In this paper, a new methodology is presented to calculate tsunami hazard at large scales. This methodology determines the tsunami flooding by using a coupled model that combines a nonlinear shallow water model (2D-H) and a volume-of-fluid model (RANS 2D-V) and applies the optimal numerical scheme in each phase of the tsunami generation-propagation-inundation process. The hybrid model has been widely applied to build a tsunami run-up database (TRD). The aim of this database is to form an interpolation domain with which to estimate the tsunami run-up of new scenarios without running a numerical simulation. The TRD was generated by simulating the propagation of parameterized tsunami waves on real non-scaled profiles. A database and hybrid numerical model were validated using real and synthetic scenarios. The new methodology provides feasible estimations of the tsunami run-up; engineers and scientists can use this methodology to address tsunami hazard at large scales.

## 1. Introduction

Recent tragic tsunami events, like those that occurred in Japan in 2011 and in the Indian Ocean in 2004, have exposed the need for further work to develop and apply tsunami risk reduction measures. The adequate evaluation of tsunami hazard in tsunami-prone areas is the first step in a proper risk evaluation (UNESCO-IOC, 2009). Determination of the tsunami hazard focuses on the estimation of the area that would be flooded during a tsunami and on the calculation of the variables or parameters that define the phenomenon in that area, e.g., wave amplitude, current depth, tsunami travel time, etc. Among these parameters, maximum run-up provides the elevation to which water from a tsunami wave will rise during its flooding process. Therefore, run-up is a key parameter that must be adequately determined when assessing the inundation of affected areas.





When tsunami hazard is addressed at a local scale (tens of kilometers or one coastal city), the optimal methodology to calculate the flooding and run-up is typically the application of validated deterministic numerical models (Álvarez-Gómez et al., 2013; Titov et al., 2011; Wang, 2009). These models allow reproduction of the 3 main tsunami processes: generation, propagation and inundation. To address these processes and to properly estimate the flooded area, high-resolution topography-bathymetry

data of the study area are required, as well as the focal parameters that define the tsunamigenic mechanism. Nevertheless, the application of tsunami numerical models has some limitations and uncertainties (Park et al., 2015; Selva et al., 2016). First, their use requires a high computational cost and expert modelers. Second, the necessary high-resolution data to properly study the hazard in local areas are not always available. In addition, the correct definition of the tsunamigenic mechanisms, e.g., the parameters of the focal mechanism, contains uncertainties in itself. Finally, even though models are evolving to reduce

uncertainties, there is still ongoing work on several aspects, such as wave transformation near the coast, interaction of waves with coastal structures, and accurate incorporation of bottom friction.

On the other hand, in large-scale studies (hundreds of kilometers or the coast of a whole country), the drawbacks of numerical models are more evident, and the lack of continuous high-resolution topobathymetry and the elevated computational cost foster

the use of other approaches. An alternative methodology to estimate the tsunami run-up and, consequently, the flooded area, includes the application of run-up analytical or empirical formulae. In these cases, numerical models, despite the lower resolution of bathymetry, adequately calculate the tsunami wave characteristics offshore and can then be used as input for the formulae. Afterwards, by applying this method to several topobathymetric profiles along the coast, the total flooded area due to tsunami action can be estimated.

The calculation and analysis of run-up was initially approached by Carrier and Greenspan (1958). They found the exact solution for the nonlinear shallow water equations for a sloping beach with non-breaking regular waves. Keller and Keller (1964) derived an analytical solution for linear shallow water waves at a constant depth moving up a constant slope beach. This geometry has become the canonical problem. Synolakis (1987) extended Carrier and Greenspan's result to this problem by

joining Carrier and Greenspan's and Keller and Keller 's solutions to provide a closed-form solution for solitary wave run-up. Synolakis' results are remarkable, as solitary waves have been widely used to model tsunamis, numerically and physically. Li and Raichlen (2001) revisited Synolakis's results to determine the importance of a higher order correction to the analytical approach. Later, Madsen et al. (2008) demonstrated that solitary waves do not represent the large scale of a tsunami, and Chan and Liu (2012) confirmed this affirmation. Madsen and Schäffer (2010) found closed-form solutions for the run-up of waves

of several shapes; their solutions included other parameters, such as the period, achieving more realistic results.

In addition, run-up has been commonly linked with the Iribarren number (Iribarren and Nogales, 1949), also called the surf similarity parameter (Battjes, 1974). Hunt (1959) joined this parameter with the non-dimensional run-up of regular waves. Kobayashi and Karjadi (1994) combined physical and numerical simulations to derive an equation to calculate run-up, using



the ratio between the run-up and the wave amplitude and its relationship with the surf-similarity parameter. Fuhrman and Madsen (2008) demonstrated that the relationship between surf-similarity and solitary waves was similar to the that between surf-similarity and period waves.

More recently, several authors have focused their work on calculating tsunami run-up by developing new models with other approaches. Sepúlveda and Liu (2016) presented expressions for the calculation of the run-up based on the parameters that defined the focal mechanism of the tsunamigenic seism. Park et al. (2015) defined the run-up for compound slopes, based on the work of Madsen and Schäffer (2010) and numerical simulations of tsunami waves on two-slope topobathymetric profiles.

However, the application of these equations and formulae is not always evident, and each approach considers different inputs. Moreover, the parameterization presented by Carrier and Greenspan (1958), extended by Synolakis (1987) and modified by Park et al. (2015), is based on theoretical bathymetric profiles. It does not explicitly consider real profiles or the geometry of the whole area, from the tsunami generation zone to the flooded area. Furthermore, the numerical models that do consider the natural geometry of the bathymetric profiles adequately predict propagation, but they cannot accurately solve the flooding
calculation, in addition to the other exposed drawbacks.

Complementing these methodologies, this work presents an alternative methodology to calculate tsunami flooding at large scales and is focused on assessing the run-up. The methodology is then applied to further develop a database from which the tsunami run-up of new scenarios can be interpolated.

The main component of the methodology is a numerical flume where the simulations are run. This flume was developed by combining a nonlinear shallow-water-equations model and a Navier-Stokes volume-of-fluid model to create a hybrid model that applies the optimal numerical scheme in each area of the flume. Time series of tsunami waves and topobathymetric profiles are used as input to calculate the run-up.

This hybrid model has been applied to further develop a database from which the run-up of new tsunami scenarios can be interpolated. This database contains an adequate representation of natural bathymetric profiles worldwide and the variability in tsunami wave shapes, allowing calculation of the tsunami run-up of new scenarios by interpolation without running a numerical simulation.

The aim of this methodology is to help specialists to further develop tsunami hazard maps at large scales, where the application of numerical models is not computationally affordable and high-resolution data are not available. This method can be used to quickly estimate the run-up in tsunami-prone areas or accurately estimate the flooded area for new tsunami scenarios.



The paper is structured as follows: Section 2 describes the developed methodology, including the parameterization of realistic bathymetric profiles and tsunami wave shapes and the construction of the numerical flume. In section 3, the application of the methodology to calculate the tsunami run-up database is discussed, together with a sensitivity analysis of the influence of each parameter on the final value of the run-up. Section 4 includes details of the tool that has been developed in order to use the

database to calculate new tsunami event run-ups. Section 5 presents the validation of the methodology with real and numerical scenarios. Finally, section 6 discusses the conclusions drawn from this work.

## 2.    Tsunami run-up hybrid model methodology

The run-up calculation methodology presented in this paper consists of the numerical simulation of tsunami waves along real non-scaled bathymetric profiles that were previously parameterized.

To carry out these simulations, a numerical flume was designed. This flume is formed by the coupling of two numerical models.

The Cornell Multi-grid Coupled Tsunami Model (COMCOT, (Wang, 2009)) solves the nonlinear shallow water equations (NLSWE) using a leap-frog finite differences scheme on a 2D horizontal domain. In addition, the IH2VOF model solves volume-averaged Reynolds-averaged Navier-Stokes (VARANS) equations based on the decomposition of the velocity and

pressure fields into mean and turbulent components using a κ-ε turbulent model on a 2D vertical domain (Lara et al., 2006). The former model is prepared to simulate the stages of tsunami propagation; meanwhile, the latter model is specially designed to simulate the coastal processes and wave transformations present when the waves reach the coastal areas.

In the flume, the strengths of both models are used to design a numerical space where tsunami waves are propagated, using

COMCOT from the deep ocean (~4 km depth) to the coast, where the capabilities of the IH2VOF model are applied to calculate the flooding. As a result, a hybrid model that adequately solves the tsunami processes in both deep and shallow waters was achieved.

Parameterized profiles and a tsunami wave time series dataset are used as input for the numerical flume. These inputs, the most

relevant aspects of the numerical flume geometry, and the coupling of the models are described below.

### 2.1  Bathymetric profile characterization

Worldwide bathymetric profiles were analyzed, with a focus on finding a parameterization that properly represents natural shapes.



To cover the existing variability in the world bathymetry, a representative sample of 50 averaged profiles was obtained from tsunami-prone coastal areas and basins, namely, the Pacific Ocean, Indian Ocean, Mediterranean Sea and Caribbean Sea (Fig. 1). Topographic and bathymetric information was obtained from the General Bathymetric Chart of the Oceans (GEBCO, International Hydrographic Organization, 2014), The European Marine Observation and Data Network (Bathymetry Consortium EMODnet, 2016) and the local bathymetry data that was available. The shape of these profiles was analyzed to perform an adequate parameterization.

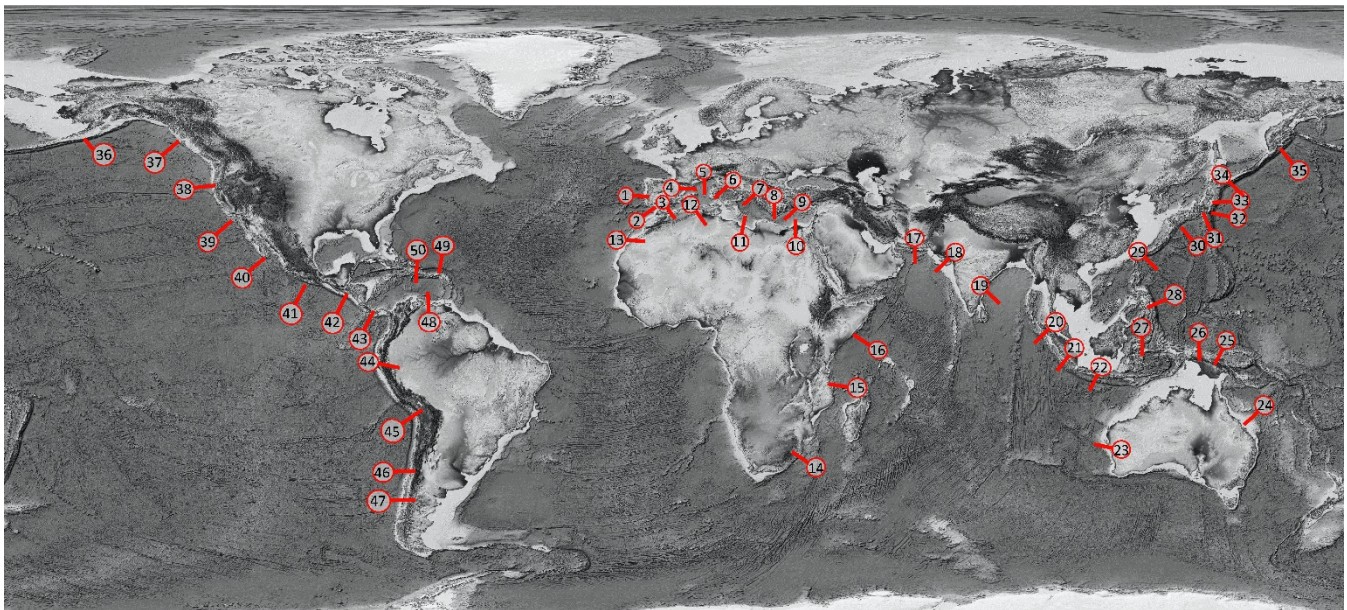

**Fig. 1. Distribution of sample profiles**

The propagation of a tsunami can affect thousands of kilometers; thus, the profiles must extend under both deep water and shallow water to capture tsunami generation to flooding. Considering this requirement, profiles were defined from inland (50 m height) to the deep ocean (~4000 m depth). To avoid singularities, each defined profile is the average profile of a 10-km-wide coastal segment. Based on the bathymetric shapes observed in this selection, the profiles were parameterized using five parameters: three slopes ($\tan \beta_1$, $\tan \beta_2$, and $\tan \beta_3$) and two depths ($d_1$ and $d_2$). Fig. 2 shows the five-parameter geometry.

As an example, in Fig. 3, a selected profile from the Indonesian coast is shown, as well as its parameterized profile that was created by applying the five-parameter geometry. The parameters for each considered profile were fitted by using a least-squares method.


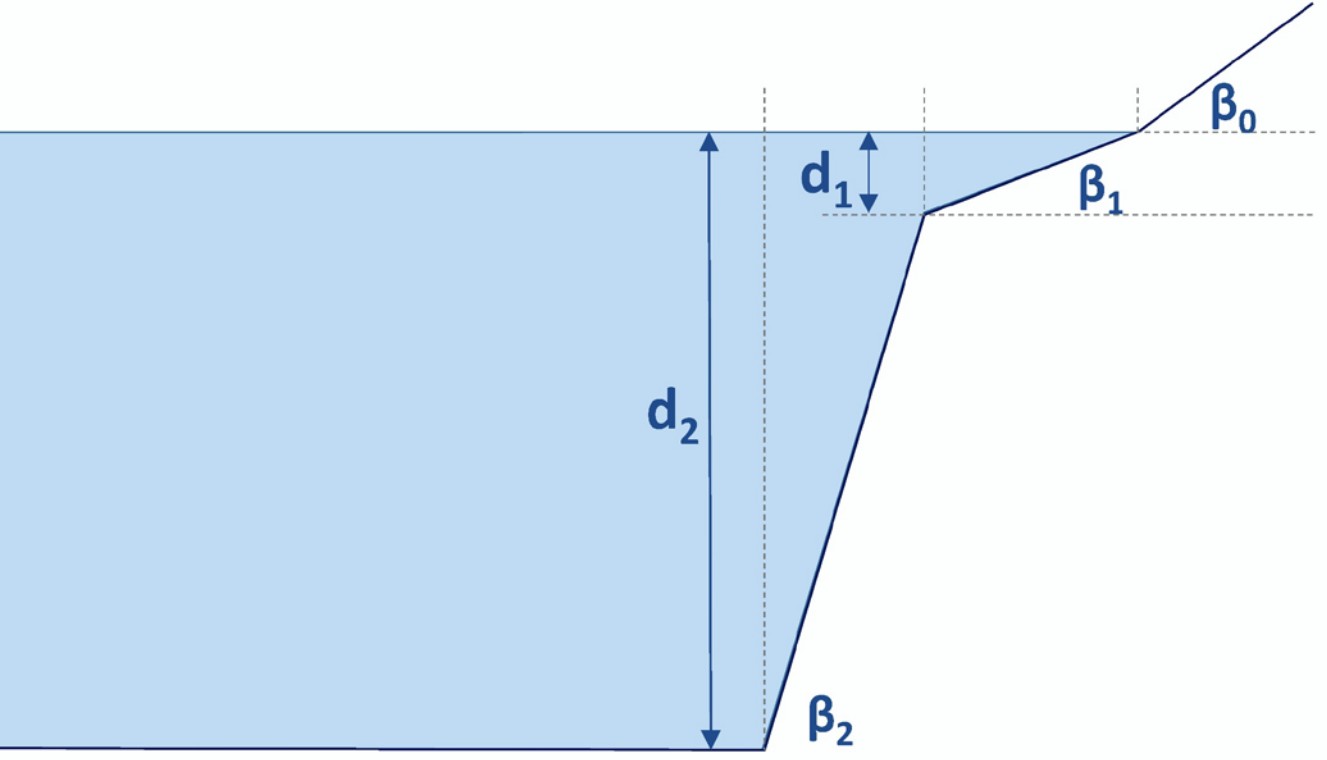

**Fig. 2. Scheme of the parameterized profiles, based on real profiles analyses. The profiles are defined by 3 angles (tan $\beta_1$, tan $\beta_2$, and tan $\beta_3$) and 2 depths ($d_1$ and $d_2$)**





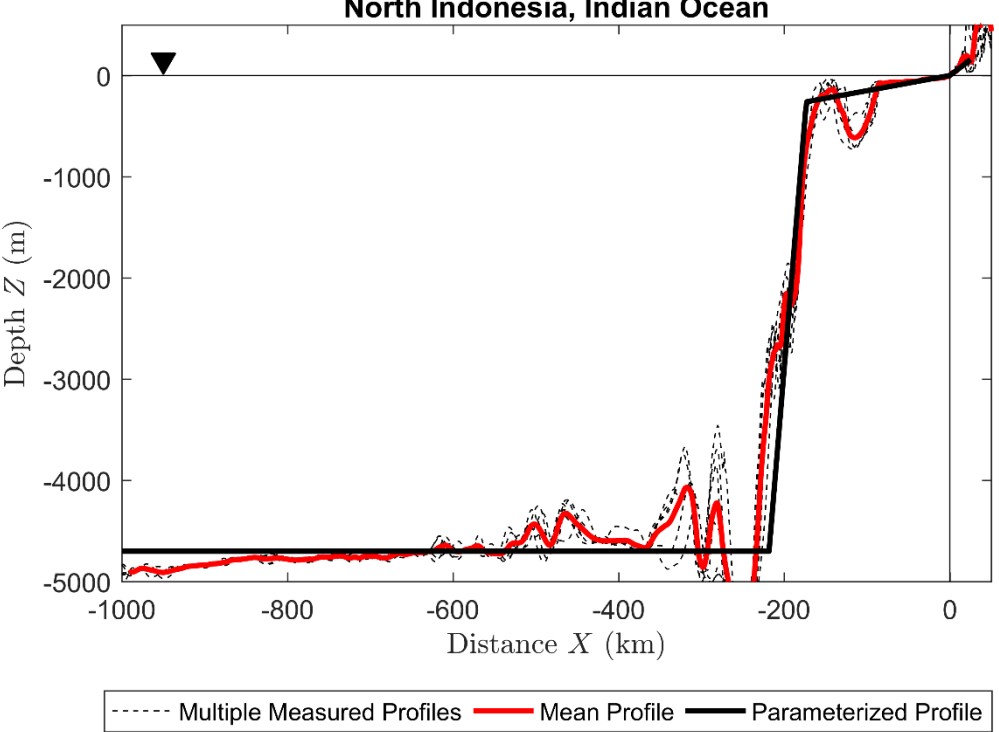

**Fig. 3. Sample of measured topobathymetric profiles on the Indonesian coast, as well as the mean and parameterized profiles. GEBCO was used as source for the topobathymetric data**

The maximum and minimum values of the 5 parameters are shown in Table 1. These values cover a wide range of the profiles

5 that can be found in nature. Despite not including all the existing geometries, the maximum and minimum values certainly provide enough information to characterize the topobathymetric profiles.

**Table 1. Maximum and minimum values of the profile parameters**

| Parameter | Min | Max |
|:---:|:---:|:---:|
| $d_1$ | 20 m | 1100 m |
| $d_2$ | 2200 m | 6000 m |
| $\tan\beta_0$ | $5.0e^{-4}$ | $1.5e^{-1}$ |
| $\tan\beta_1$ | $5.0e^{-4}$ | $2.5e^{-2}$ |
| $\tan\beta_2$ | $1.0e^{-2}$ | $2.0e^{-1}$ |





## 2.2 Initial tsunami wave characterization

The numerical flume described in detail in the next subsection requires not only the topo-bathymetric profile characterization but also the characteristics of the tsunami waves as input. To use these data as input for the hybrid model, a time series of the offshore wave amplitude must be provided. These time series could be obtained from either records of real tsunamis, e.g., from

DART buoys, or from the results of numerical model tsunami propagation. In this case, COMCOT (Wang, 2009) was adopted. This model calculates all stages of tsunami modeling (generation, propagation and coastal flooding).

The generation of the tsunamis in COMCOT is approached via elastic finite fault plane theory, using the so-called Okada model (Okada, 1985). This model assumes an idealized rectangular fault plane as a representation of two colliding tectonic plates. The Okada model requires 7 focal mechanism parameters as input to calculate the initial deformation of the water

surface due to the earthquake. These parameters are the focal depth ($h_{focal}$), rupture length ($L$) and width ($W$) of the fault plane, dislocation ($D$), strike direction ($\theta$), dip angle ($\delta$) and slip (rake) angle ($\lambda$). A simulation of the numerical model provides the wave amplitude time series to be used as input for the hybrid model.

## 2.3 Numerical flume geometry

The dimensions of the numerical flume vary with the profile characteristics, adapting the domain for each simulation. The

geometry of the flume is shown in Fig. 4. The total length $L$ of the flume is split in two components: $L_{off}$ is the submerged part of the profile and $L_i$ is the inland part of the profile.





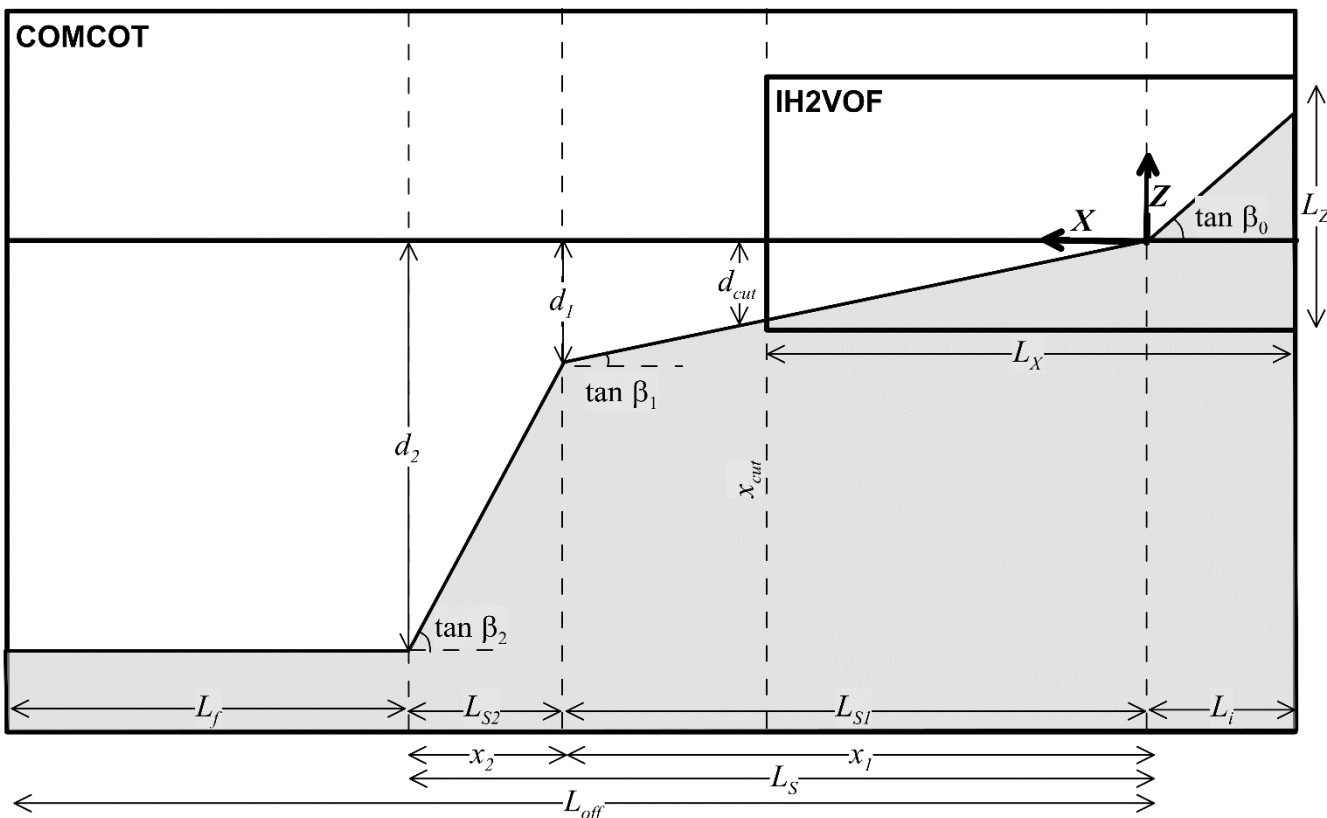

**Fig. 4.** Numerical flume geometry, including the 5 parameters that define each profile (tan $\beta_1$, tan $\beta_2$, tan $\beta_3$, $d_1$ and $d_2$) and the general location of $x_{cut}$, where numerical models are coupled

5  $L$ is determined for each simulation according to the profile parameters (tan $\beta_1$, tan $\beta_2$, tan $\beta_3$, $d_1$ and $d_2$), and the tsunami wave

length is $\lambda = T \cdot \sqrt{g \cdot h}$, where $T$ and $H$ are the tsunami wave period and height and $g$ is the gravitational acceleration:

$$L = L_i + L_{off}$$

$$L_i = \frac{1}{50} \cdot tan\beta_0$$

$$L_{off} = L_f + x_2$$

$$L_f = \left\lceil \frac{1.2 \cdot \lambda}{10 \cdot \Delta x} \right\rceil \cdot \Delta x$$

$$x_2 = \frac{d_1}{tan\beta_1} + \frac{d_2 - d_1}{tan\beta_2}$$





where $\Delta x$ is the resolution (cell size) of the simulation with the COMCOT numerical model, as described in detail in the next section.

The IH2VOF domain is located in the shallowest part of the profile, with a sufficient area of the inland domain to obtain an
accurate measurement of run-up and an area as long as possible for the wave propagation.

### 2.4  Numerical models coupling

The coupling of the numerical models was focused on accurately locating the border position between the models, $x_{cut}$ (see Fig. 4). This location is optimized in the domain of the IH2VOF model for every tsunami scenario, since that area is the most computationally demanding. Two criteria are followed for this optimization: 1) maximize the area of the IH2VOF domain and
2) simultaneously ensure that the flooding does not exceed the inland end of the IH2VOF domain.

To achieve this optimization, it is necessary to know a rough value of the run-up in advance in order to fit the inland part of the grid. In this sense, the more accurate the rough estimate of the run-up is, the fewer inland cells are wasted (without flooding), meaning that the IH2VOF performance is optimized. Clearly, the complete run-up must be fully covered by this
model domain, meaning that the vertical length of the onshore grid $L_z$ must be adequate. To determine this horizontal length in advance, each simulation is precalculated with only COMCOT to obtain an approximation of the run-up of the considered tsunami scenario.
The transference of data between models occurs at $x_{cut}$. IH2VOF requires as input a time series of sea surface deformation and a velocity profile; hence, these data are obtained as an output of the COMCOT model. Nevertheless, in most cases, the tsunami
wave length $\lambda$ is considerably longer than the length of the IH2VOF grid $L_x$ ($\lambda > L_x$). Therefore, before the entire wave has passed $x_{cut}$, the reflected wave has already reached back to that point; therefore, the amplitude tsunami wave series from COMCOT used in IH2VOF along $x_{cut}$ would be "contaminated" with the reflected wave.

To avoid this situation, a second simulation with only COMCOT is performed for the considered scenario. In this simulation,
the topobathymetric profile is the same, but $\beta_0$ is set to 0 from $x_{cut}$, and the right inshore boundary is left open (see Fig. 5). This approach minimizes the influence of the reflection, allowing the input data that COMCOT transfers to IH2VOF to be accurately obtained at the $x_{cut}$ position.



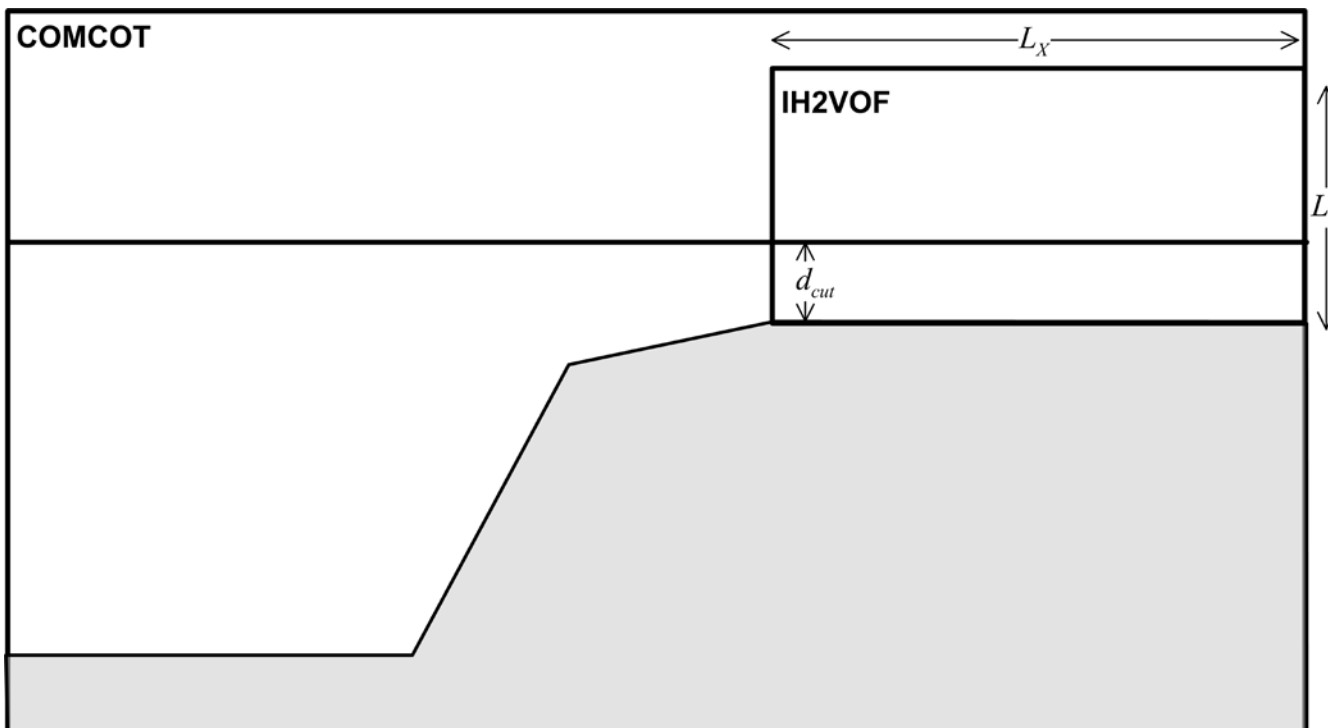

**Fig. 5. Numerical flume geometry with a modified profile to avoid the reflection phenomena on the x$_{cut}$ position**

To attest the effectiveness of this approach, an example of the wave height propagation of a tsunami wave with *H*=5.6 m for

5  *T*=40 minutes in the numerical flume is shown in Fig. 6. Fig. 6a shows the propagation of the wave height in the unaltered flume with the reflection effects, and Fig. 6b shows the same propagation in the modified flume, in which the reflection effects are minimized. In the plots in Fig. 6, the x-axis is along the length of the flume, the y-axis is the time of the simulation, and the x$_{cut}$ position is marked as a red line. The example wave enters the flume after 10 minutes (600 seconds) of simulation and propagates towards the coast (zero on the x-axis), reaching the x$_{cut}$ position after 30 minutes (1800 s) of simulation. At the x$_{cut}$

10  position, a reflected wave on the order of 1 m height is reduced by 95%, making it possible to obtain the boundary conditions for the IH2VOF simulation.







**Fig. 6.** Propagation of a tsunami wave in the numerical flume on the distance-time plane. $x_{cut}$ position is indicated on the plane with a red line. a) In the initial domain, a reflected wave is observed, aliasing the boundary condition for the IH2VOF domain. b) The modified domain, in which a constant infinite depth is modeled after the $x_{cut}$ position. In this case, the reflection of the wave is eliminated, allowing the boundary conditions for the IH2VOF model to be obtained

Therefore, to sum up, the procedure to simulate the propagation of a tsunami wave in the numerical flume follows the following steps: i) COMCOT domain design based on the profile parameters and tsunami wave; ii) COMCOT simulation to obtain a first estimate of the expected run-up; iii) design of the IH2VOF domain, based on the run-up estimation; iv) calculation of the position of $x_{cut}$; v) design of the COMCOT domain inland with a modified profile to eliminate the effect of the reflected wave;

vi) COMCOT simulation to obtain the boundary conditions (input) for the IH2VOF simulation; vii) IH2VOF simulation and viii) run-up determination in the IH2VOF domain.

To assess the tsunami hazard in a tsunami-prone area, this model can be applied to several profiles all along the studied coastal area. The methodology provides the run-up at each of the profiles, allowing the flooded area to be estimated as an envelope of

the run-up limits.

## 3.   Application example: further development of a tsunami run-up database

The presented hybrid model was created with the aim of applying it to generate a tsunami run-up database (TRD) from a combination of bathymetric profiles and tsunami waves. The objective of this database is to create an interpolation space that

allows the instant evaluation of the tsunami run-up, without needing to run numerical simulations.

This database contains the run-up of tsunami scenarios that are combinations of parameterized tsunami waves and parameterized bathymetric profiles. These scenarios have been simulated within the described numerical flume.

The following section is focused on explaining the details of the development of the TRD, the selection of the bathymetric profiles and tsunami waves, and the simulations run with the hybrid numerical model. Finally, an interpolation tool, which

was developed ad hoc to apply the database to the instantaneous estimation of the run-up, is presented.

### 3.1   TRD bathymetric profiles

The parameterization of the 50 bathymetric profiles that were selected worldwide (see Fig. 1 and Table 1) were added to the TRD. The 5 parameters of each of these 50 measured profiles were obtained by means of a least-squares fitting method.

To increase the number of cases included in the TRD, more realistic profiles were added as combinations of the 5 parameters. To generate these profiles, the ranges of the values of each parameter over the 50-profile sample set were analyzed, with a



focus on identifying trends or rules that characterize their variability. The new profiles follow these trends, avoiding the inclusion of unrealistic combinations of parameters (e.g., $(d_2 - d_1)$ was always shorter than 2200 m and $x_1$ was always shorter than 210 km). By using these realistic combinations of parameters, the TRD was expanded to 5000 profiles.

5    Finally, from those 5000 profiles, a selection of 49 profiles was made by means of the maximum dissimilarity algorithm (Camus et al., 2011). These 49 profiles assure a maximum variability in the profiles to further develop the TRD. The parameters of the 49 chosen profiles are given in Table A1 of Annex A.

### 3.2  TRD tsunami wave parameterization

The tsunami waves for the TRD were obtained by means of simulations of realistic scenarios using the COMCOT model. Based on these simulations, the tsunami waves were characterized by 2 parameters: tsunami wave height ($H$) and period ($T$) at the depth $d_2$.

To generate the tsunami wave shapes with COMCOT, an infinite horizontal domain with a constant water depth was used. In the analyses of the seven focal mechanism parameters (see 2.2), some simplifications were assumed. First, to generate higher tsunami waves, the three angles were fixed to the combination that provides the maximum tsunami height, i.e., θ=0°, δ=90° and λ=90°. Second, $D$ was defined in Table 2, and a width of $W = (M_0/6.25\mu\gamma)^{1/3}$ is obtained by assuming a rectangular fault with proportion $L/W = 2.5$ and using the $M_0$ formulation (Table 2). Finally, Kanamori and Anderson (1977) provided a

relationship between the seismic moment and earthquake magnitude: $M_w = 2/3 \cdot \log(M_{o)} - 6.07$.

**Table 2. Formulations used in the definition of the parameters of the tsunami waves included in the addition to the database**

| Formulation | Definitions | Source |
|---|---|---|
| $D=\gamma L$ | $\gamma$=6.5×10$^{-5}$ | Scholz and Harris (2002) |
| $M_o=\mu SD$ | $S=LW$; $\mu$=2.5×10$^{11}$ Aki (1972) | Steketee (1958); Burridge, R. and Knopoff (1967) |

Taking these parameters into account, the tsunami waves can be obtained in terms of earthquake magnitude ($M_w$), focal depth

($h_{focal}$) and water depth ($d$). Therefore, the influence of these three parameters on the tsunami wave height and period was explored and depicted in a general scheme in Fig. 7. As it could be intuitively expected, the higher the magnitude of the earthquake is, the higher the tsunami wave height; however, the deeper the focal depth is, the lower the tsunami wave height. On the other hand, regarding the tsunami wave period, the period increases with the earthquake magnitude or focal depth. The water depth in the rupture area affects only the tsunami period, which increases when this depth decreases.


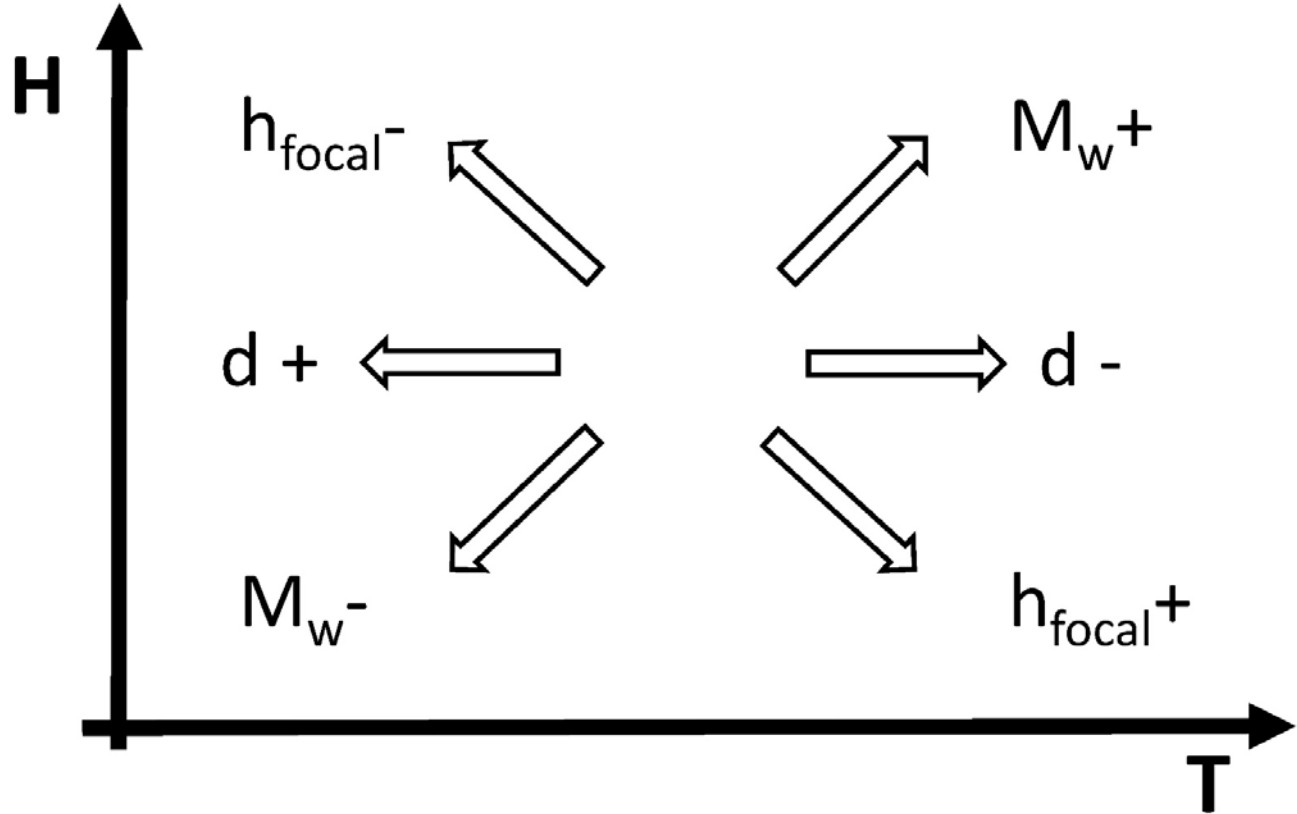

**Fig. 7.** General scheme of the tsunami wave height (*H*) and period (*T*) behavior in relation to the water depth (*d*) in the generation area and earthquake focal depth ($h_{focal}$) and magnitude ($M_w$)

Following this characterization, a set of tsunami waves was selected, covering period values from 5 to 40 minutes and wave height values in the source area (depth $d_2$) from 0.2 to 2.0 m. In generation areas, tsunami waves are commonly within these ranges (Papadopoulos, 2016). The considered waves are peak or positive waves, meaning that the wave height was considered from the sea level. The tsunami wave heights (from A to K) and periods are given in Table 3. The interpolation is limited to the H and T ranges included in the database. The incorporation of new waves will complement the existing wave database and increase the range of application of the methodology.

**Table 3. Tsunami wave height and period at the source area for the current events included in the TRD**

| Name | Height (m) | Period (mins) |
|:---:|:---:|:---:|
| A | 1.6 | 35 |
| B | 0.5 | 35 |





| | | |
|---|---|---|
| **C** | 0.5 | 8 |
| **D** | 0.2 | 5 |
| **E** | 1.5 | 15 |
| **F** | 0.5 | 15 |
| **G** | 1.5 | 25 |
| **H** | 0.5 | 25 |
| **I** | 1.0 | 35 |
| **J** | 1.0 | 15 |
| **K** | 1.0 | 25 |

## 4.  Run-up estimation by interpolation of the TRD

The procedure explained at the end of section 2.4 was followed to calculate the run-up of the combination of the tsunami waves and the 49 topobathymetric profiles. Therefore, the TRD is increased by 539 scenarios, provided by 7 parameters ($tan\beta_0$, $tan\beta_1$, $tan\beta_2$, $d_1$, $d_2$, $H$ and $T$). These simulated scenarios constitute the 7-dimension interpolation domain in which new run-up calculations are carried out. The interpolation procedure and the result of its application are described next. A tool to calculate the run-up by interpolation was developed. This tool allows the simultaneous analysis of the influence of each parameter on the final value of the run-up.

### 4.1  The interpolation tool (IH-TRUST)

To interpolate the value of the run-up for new scenarios, a numerical tool was programmed. This tool, called *Instituto Hidráulica-Tsunami Run-Up Simulation Tool* (IH-TRUST), processes the profile and wave data to calculate the run-up, and it performs an interpolation of the 7 parameters considered in the TRD. IH-TRUST consists of three modules, or elements (Fig. 8).



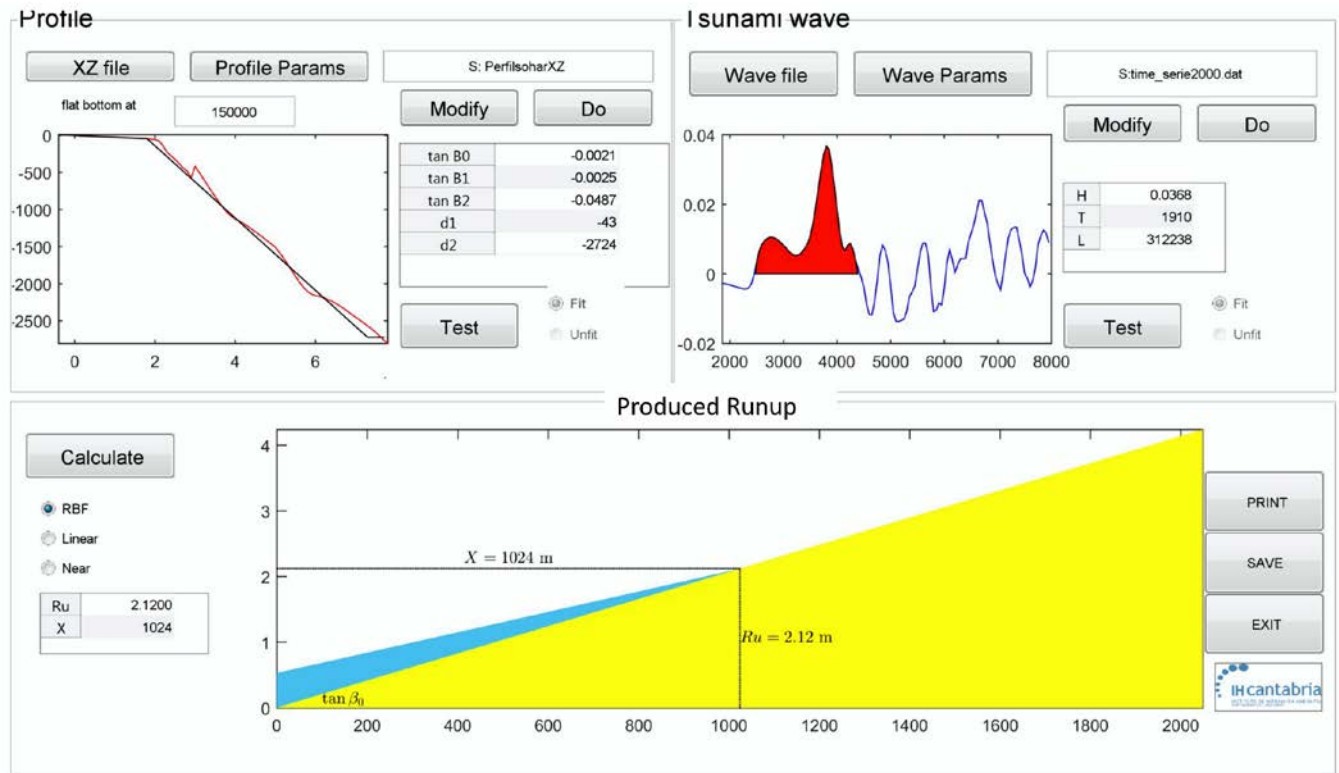

**Fig. 8. IH-TRUST interface, showing its 3 elements: the bathymetric profile, the tsunami wave parameterization and the run-up calculation**

5    In the first element, the tool calculates the parameterization of the real topobathymetric profile into 5 parameters: $tan\beta_0$, $tan\beta_1$, $tan\beta_2$, $d_1$, and $d_2$. This parameterization is approached by means of a least-squares fitting. An example of the fitting of a profile is shown in Fig. 9, in which the main plot shows the quadratic error in terms of distances $X_1$ and $X_2$, and the star indicates the minimum error, which is consequently the position of the best set of five parameters. The subplot shows the original profile and the parameterized profile. Afterwards, the tool verifies that the parameterized profile is included in the ranges of the

10    parameter values contained in the interpolation space.



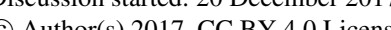

**Fig. 9. Parameterization of the bathymetric profiles. The figure maps the $E_{rms}$ values as a function of the distance from the coast to $d_1$ ($X_1$) and the distance from the coast to $d_2$ ($X_2$) obtained during the process of finding the best parameters for a profile. In the subplot, the original profile and the parameterization are shown**

5    Fig. 10 shows a representation of all the profile domains in black and the introduced profile in red. A set of bars indicates the acceptable values for each parameter, and a star marks the position of each parameter for the new profile.





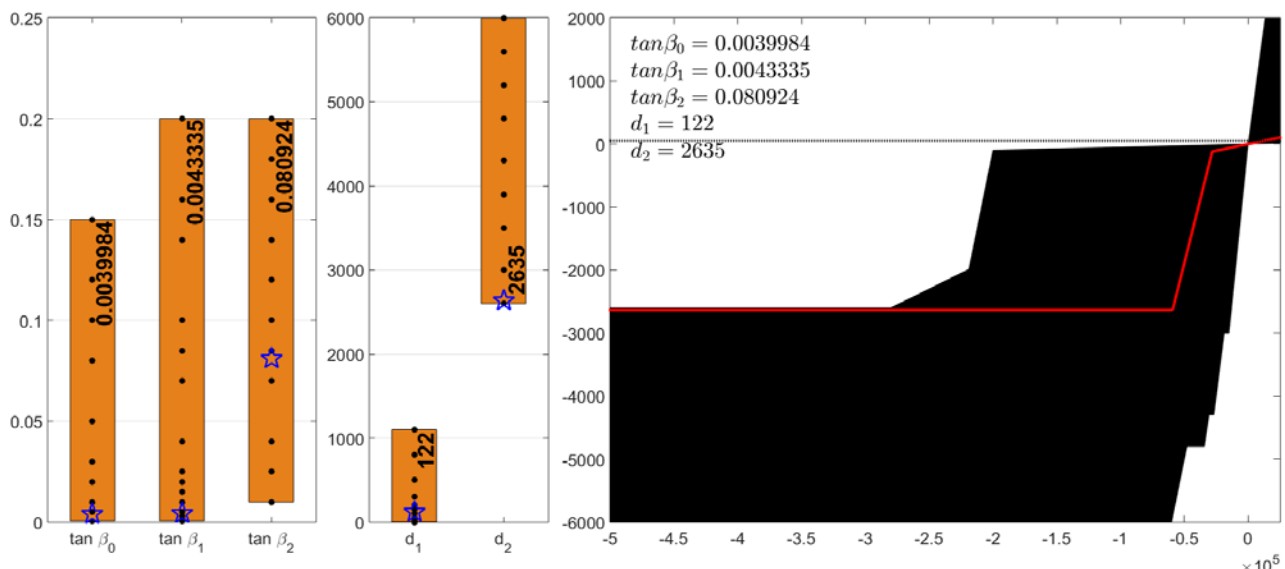

**Fig. 10. Fitting of a topobathymetric profile in the TRD. Each parameter value (left) and parameterized profile shape (red) for the profiles are included in the TRD (black)**

5   In the second element, IH-TRUST calculates the values of the $H$ and $T$ of the tsunami wave to be assessed at depth $d_2$. The tool reads a time series containing the tsunami shape and calculates $H$ and $T$. $T$ corresponds to the time between the first two zero crossings for positive heights, and $H$ is the maximum wave height observed within period $T$, but the tool allows to manually set the period within the time series, if desired (Fig. 11).





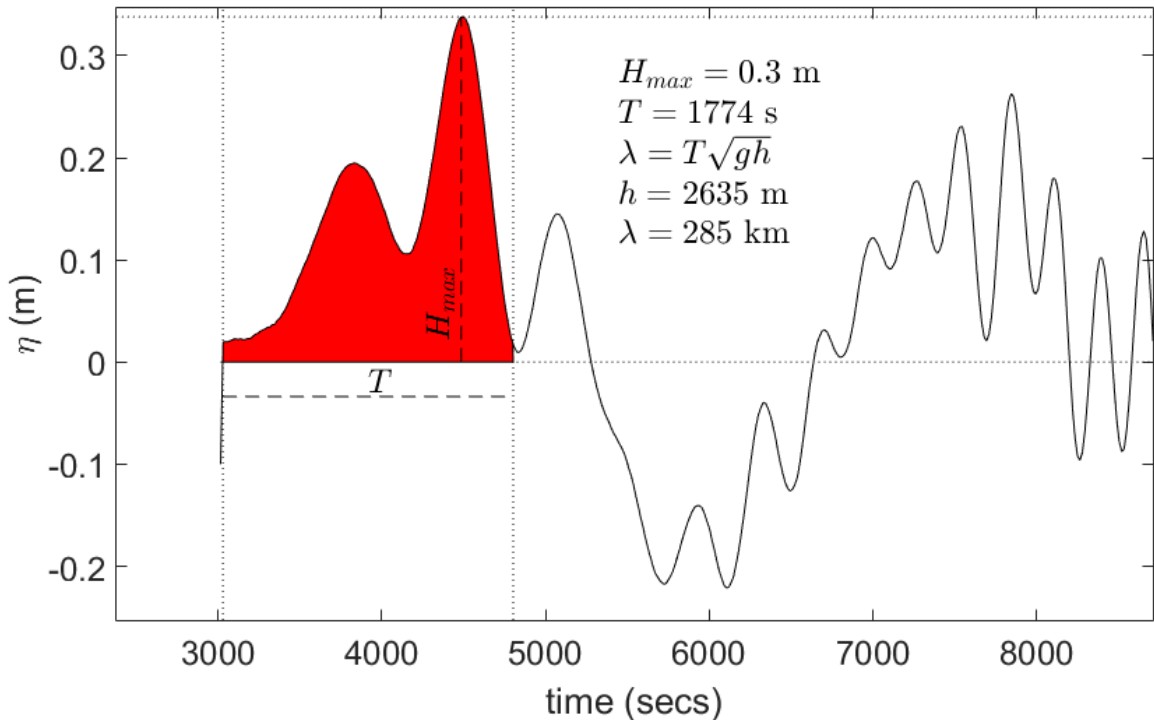

**Fig. 11.** **Tsunami wave profile parameterization in the IH-TRUST tool, including the part of the tsunami time series considered (in red) to calculate the period *T* and the height *H***

After the wave parameters are calculated, IH-TRUST checks if the tsunami wave fits in the interpolation domain of the
database. Fig. 12 shows the tsunami waves included on the database, the area where the interpolation is valid and the position
of the tsunami wave that is being studied.





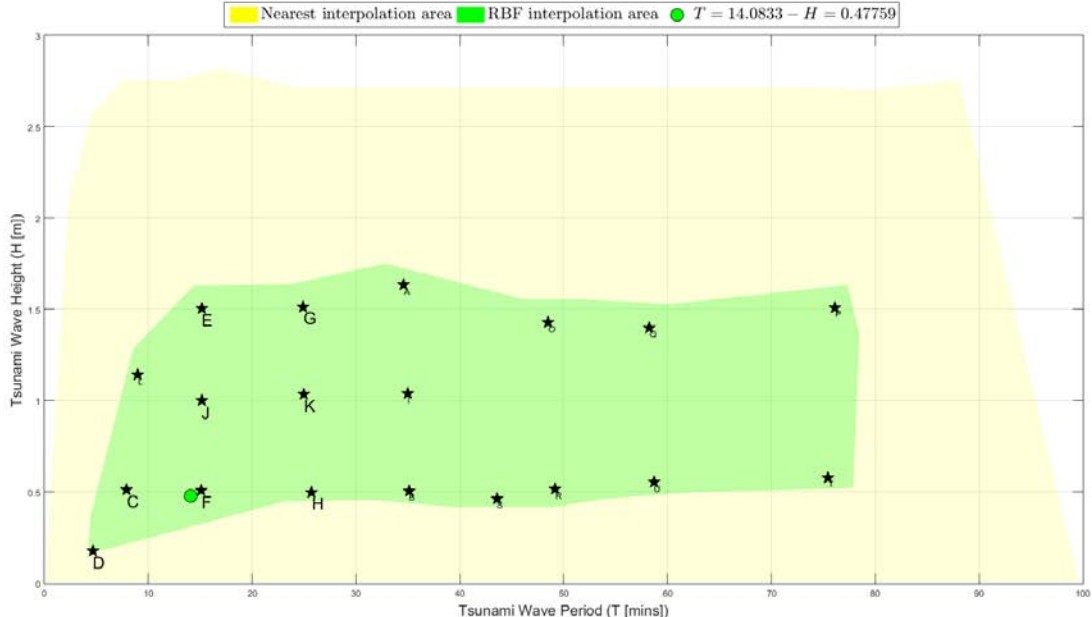

**Fig. 12. Tsunami wave height and period cases included in the database. The point corresponding to any new wave should fall in the green shadow in order for it to be able to be interpolated with the generated TRD**

5    Finally, in the third element of IH-TRUST, the results of the calculation of the run-up *Ru* is given based on the profile parameters and the tsunami wave. The interpolation (Fig. 8) is calculated by means of the RBF (Camus et al., 2011) and linear and nearest interpolation methods. In addition, the horizontal flooding distance *X* is calculated using the inland slope. The tool uses an RBF interpolation by default, but the nearest or linear methods are also available, since they are useful to calculate events that plot closer to the boundary of the valid interpolation area.

### 4.2  Influence of the profile parameters on the tsunami run-up

The TRD and IH-TRUST were used to explore and analyze the influence of each parameter of the profile on the final value of the run-up. This analysis was approached by evaluating scenarios in the TRD. Although it is out of the scope of this paper, to understand the influence of each parameter of the bathymetric profile, several tests were conducted with a mean profile

15   ($tan\beta_{0m}$=0.080, $tan\beta_{1m}$=0.09, $tan\beta_{2m}$=0.110, $d_{1m}$=500, and $d_{2m}$=4350) and by varying only one of the 5 parameters that define the profile at a time; additionally, several values of *H* and *T* inside the boundaries of the domain were considered.





For each pair of values of *H* and *T*, 4 of the 5 profile parameters were kept constant, and the run-ups were calculated using IH-TRUST with the TRD by varying the 5th parameter.

The effect of the variation in *Ru/H* as a function of the parameters are shown in Fig. 13 and Fig. 14.







**Fig. 13. Maximum run-up (normalized by wave height) as a function of the profile slopes (*tanβ₀*, *tanβ₁*, and *tanβ₂*) of the parameterized profiles for different wave heights.**

The continental slope effect (Fig. 13), parameterized as $tanβ_2$, produces a maximum $Ru$ when $tanβ_2$ is close to $tanβ_1$,
5  reproducing a single slope profile. For smaller $tanβ_2$ values, $Ru$ decreases rapidly due to wave shoaling. Low values of $tanβ_2$ also indicate a large platform with a low slope, where the shoaling increases the wave height and the wave energy diminishes gradually due to bottom friction until wave breaking occurs. Thus, the energy flux that reaches the shore decreases with the run-up height. The profile typology characterized by a low value of $tanβ2$ is closer to Synolakis's canonical problem.

**Fig. 14. Maximum run-up heights as a function of the profile slopes (*d₁* and *d₂*) of the parameterized profiles for different wave heights**





The higher the $tan\beta_2$ is, the shorter the platform, reducing the energy dissipation and allowing the slopes to have similar $tan\beta_0$ and $tan\beta_1$; this maximizes the run-up height.

Regarding $tan\beta_1$, when $d_1$ is constant (Fig. 13), the higher $tan\beta_1$ is, the shorter the length of the shelf, reducing tsunami wave

shoaling. In this case, the wave steepness increase drastically near the coast and breaks abruptly, triggering a considerable dissipation of energy within a short length; this effect reduces both the energy flux on the coast and the run-up.

Finally, the influence of $tan\beta_0$ on the final value of the run-up is less important than those of $tan\beta_1$ and $tan\beta_2$. The run-up decreases as $tan\beta_0$ increases. Due to the effect of gravity, the flow ascends less if greater slopes are present. This aspect is

strengthened by the reflection of the energy.

The behavior described above can be classified into four types of profiles, in terms of $tan\beta_1$ and $tan\beta_2$ for a constant $d_1$, shown in Fig. 15. Profile types A) and B) have a $tan\beta_2$ lower and higher than $tan\beta_1$, respectively, and profile types C) and D) have a fixed $tan\beta_2$ and $tan\beta_1$ is lower and higher than $tan\beta_2$, respectively. Of these four cases, the maximum $Ru$ is observed when

$tan\beta_2$ is approximately $tan\beta_1$, but important differences are observed because the slopes differ. In profile types A) and D), the wave dissipation is important to reducing the wave amplitude and consequently the final run-up. Otherwise, B) and C) probably reflect typical profiles, in which shoaling is the most important process affecting the wave propagation.

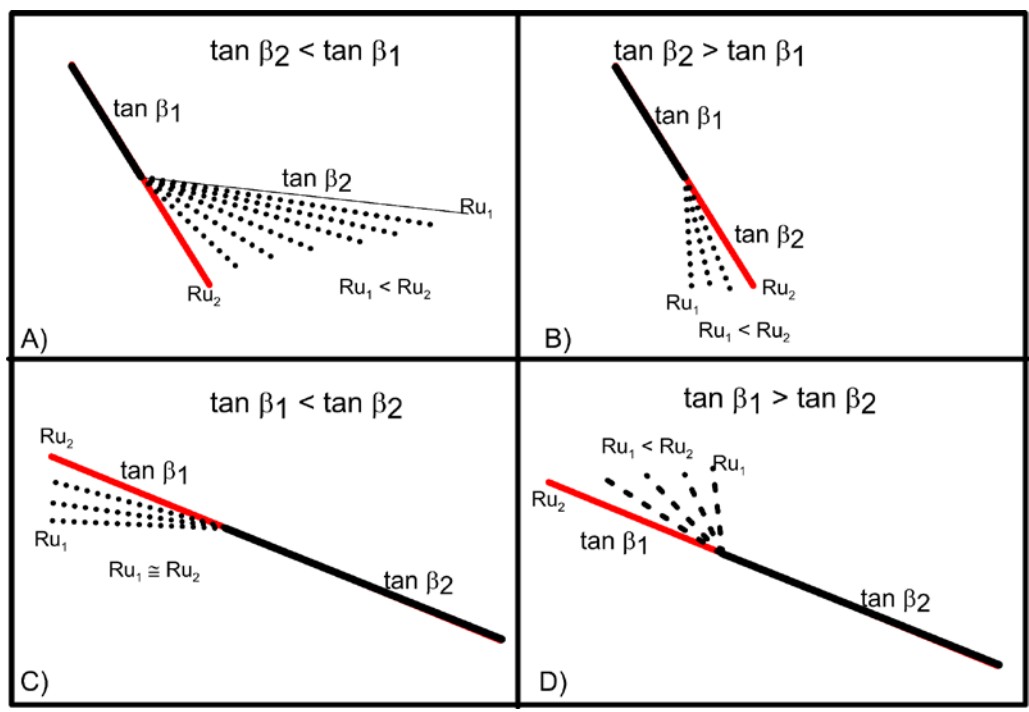





**Fig. 15.** **The four types of possible profiles: a)** *tanβ₂ < tanβ₁*; **b)** *tanβ₂ > tanβ₁*; **c)** *tanβ₁ < tanβ₂*; **and d)** *tanβ₁ > tanβ₂*. **Discontinuous lines indicate possible profiles for each type**

The influence of depths $d_1$ and $d_2$ is shown in Fig. 14. For deeper continental shelf depths $d_1$, the shelf is wider and, consequently, the bottom friction affects the wave over a longer profile, creating a run-up smaller. For a constant $tan\beta_1$, lower values of $d_1$ represent a shorter continental shelf, and abrupt and dissipating wave breaking. Moderate values of $d_1$ are characterized by a gradual tsunami wave shoaling, during which the bottom friction allows a maximum run-up. From that critical point, higher values of $d_1$ mean a longer continental shelf, generating a larger frictional area, reducing the energy flux that reaches the shore and consequently diminishing the run-up.

In Fig. 14b, it can be observed how the run-up increases almost linearly with $d_2$. The effect of $d_2$ in the run-up is similar to the effect of $tan\beta_2$. The shallower $d_2$ is, the greater the shoaling and the higher the wave. The wave energy diminishes gradually due to bottom friction until wave breaking, which depends on the tsunami wave height. In addition, it was found that although the variations in wave height produce different $Ru/H$ values for the same profile, the influence of the variation in the wave period is negligible. Therefore, different wave heights but not different periods are shown in Fig. 13 and Fig. 14.

Finally, these results highlight the importance of using an accurate geometry to define the run-up. The influence of $d_2$ and $tan\beta_2$ in the final run-up estimation is considerable, and the use of complete profiles, from the generation area to the coast, is necessary but not considered in traditional approaches and simplifications.

## 5. Validation of the methodology with numerical test results and observational data

The methodology presented here aims to calculate the tsunami run-up in coastal areas. This calculation can be applied to study the run-up of historic events but also to calculate the run-up of potential scenarios, which are the primary focus. These potential cases are used to evaluate tsunami hazard and the flooded area when a tsunami occurs. As mentioned in the introduction, run-up is commonly assessed by means of numerical simulations.

Therefore, to validate this methodology/tool, the results of its application have been compared with both high-resolution numerical simulations of potential events and historical tsunami run-up scenarios.

The results of these comparisons are detailed in the following subsection, which is focused on describing the strengths and limitations of the methodology for each case.





### 5.1 Validation with numerical model simulations

This validation was carried out as follows: first, a topobathymetric profile of the study area was obtained using the GEBCO database. On that profile, a point was selected offshore, and the time series of the tsunami was extracted at that point from the COMCOT numerical simulation of the event. Using the topobathymetric profile and the time series as input for the IH-TRUST tool, the run-up was interpolated by using the created database. The interpolated run-up was then compared to the run-up obtained by using the high-resolution numerical simulation of the potential scenario.

Three numerically simulated scenarios with high resolutions have been selected for the validation. All these scenarios are from real projects, studies and published papers that were focused on analyzing and assessing the tsunami hazard in coastal areas worldwide and characterizing the potential flooded areas due to tsunami events in the selected zones. These simulations used high-resolution topographic and bathymetric data to construct grids with 30 m cells.

#### 5.1.1 Tsunami scenario in Trujillo, Peru

The results of the application of the methodology were compared to the results of a high-resolution numerical simulation of a magnitude 8.5 event in the subduction zone located along the coast of Trujillo, a municipality in northern Peru. This synthetic scenario represents the event that occurred in this zone in 1619 and is part of the study *Probabilistic evaluation of the hazard and vulnerability under natural disasters in the metropolitan area of Trujillo*, funded by the Inter-American Development Bank (IHCantabria, 2013). The numerical simulation used a 30-m-resolution grid to accurately calculate the flooded area for a tsunami wave height and period of approximately 1.5 m and 400 s at a depth of 3000 m.





**Fig. 16. Flooded area in the municipality of Trujillo in Peru, due to a tsunami triggered by an 8.5 magnitude earthquake, including 3 selected profiles with the run-up obtained by using the numerical model. The coordinates of the exact locations where the run-up was estimated are provided in Table 4**

In Fig. 16, the flooded area map of Trujillo, as well as the selected profiles, are shown. In the study, the numerically calculated run-ups at those profiles (Fig. 17) were 8.9, 10.6 and 12.8 m. The corresponding values for the run-up obtained by interpolating the TRD with the IH-TRUST tool were 8.8, 10.5 and 11.6 m (see Table 4). Compared to the results of the numerical simulation, these 3 values from the 3 zones of the study area provide a good approximation of the tsunami flooding.



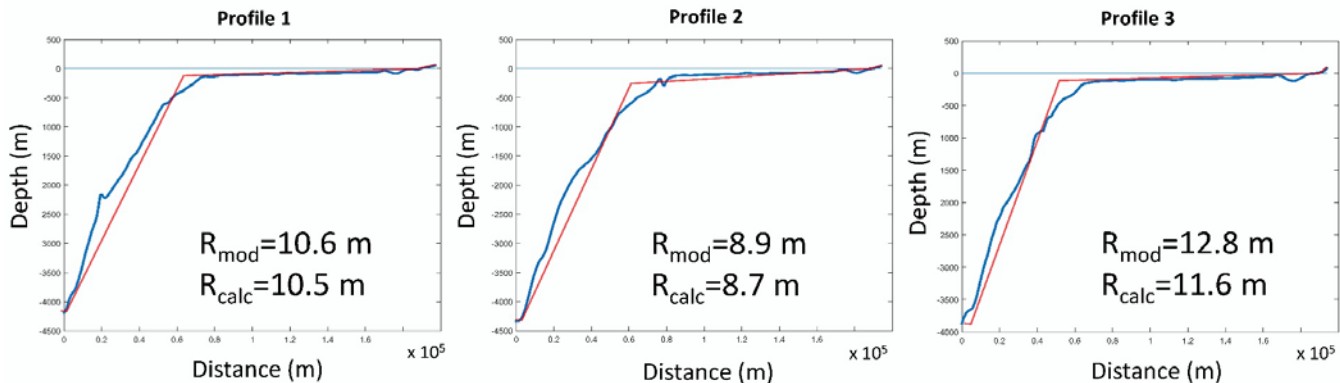

**Fig. 17. Topobathymetric profiles selected in Trujillo, Peru to validate the methodology. The topobathymetric profile (blue) and the parameterized profile (red) are compared**

### 5.1.2 Tsunami scenario in La Libertad, El Salvador

5 Following the same procedure, a validation case was addressed in El Salvador. The event is a potential scenario of an earthquake of magnitude 8.1 along the El Salvador thrust, which is in the subduction zone along the El Salvador coast. The study area is the flat area of La Libertad, on the western side of this Central American country. This high-resolution numerical simulation is part of the project *Tsunami Risk Assessment in El Salvador*, financed by AECID (Spanish Agency for International Cooperation and Development) during the period 2009–2012 (Álvarez-Gómez et al., 2013). The resolution of

10 the numerical simulation was 30 m, and the grid that was built for the propagation and inundation calculations used data from local bathymetric campaigns and high-resolution topographic studies. The tsunami wave height and period at a depth of 3000 m were approximately 0.9 m and 700 s.



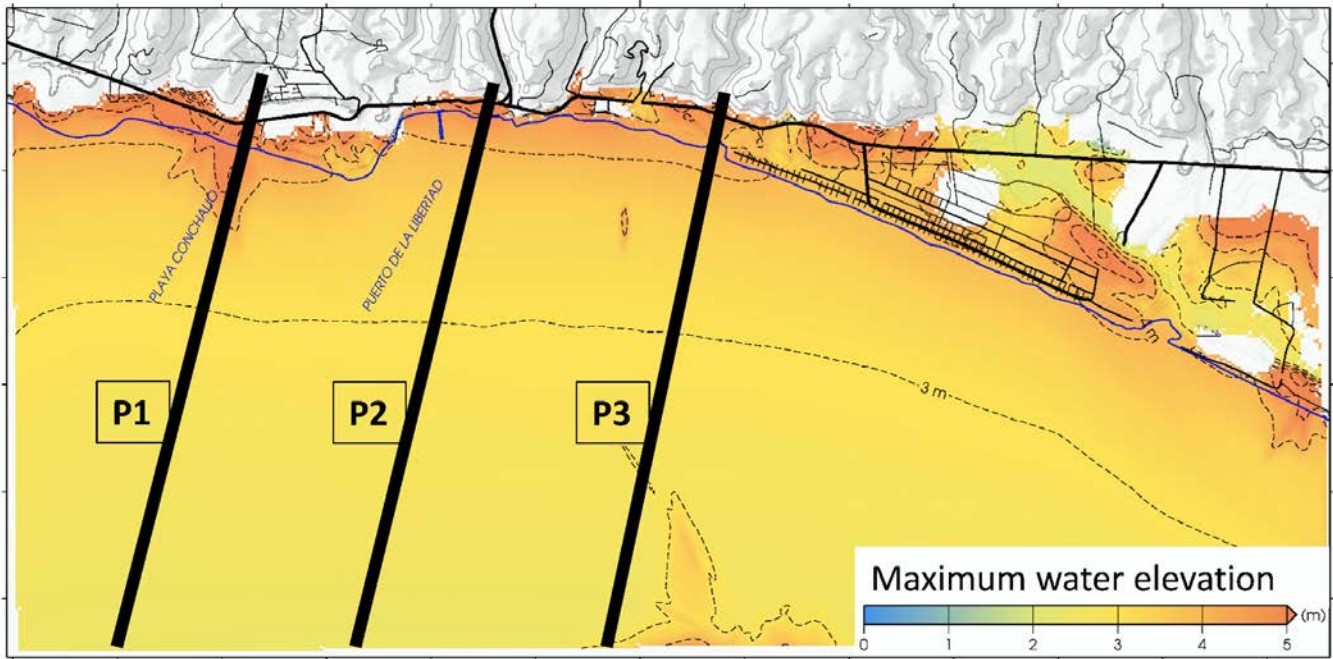

**Fig. 18. Flooded area in the municipality of La Libertad in El Salvador, due to an 8.1 magnitude event with epicenter along the coast of this Central American country. The exact locations where the run-up was estimated are provided in Table 4**

In Fig. 18, the flooding map that was part of this project is shown, and in the same figure, the selected profiles have been superimposed. In this simulation, the run-ups obtained at the three profiles in Fig. 18 were 5.2, 5.5, and 6.3 m. The corresponding run-ups obtained by interpolating the TRD with the IH-TRUST tool were 6.2, 6.1 and 7 m.

### 5.1.3     Tsunami scenario in Muscat, Oman

As part of the Multi Hazard Risk Assessment System of Oman (Aniel-Quiroga et al., 2015), more than 3000 potential tsunami
events were numerically modeled. A selection of these events were selected to assess the tsunami hazard for some specific municipalities in Oman by means of high-resolution numerical simulations of the generation, propagation and inundation processes, with a 30 m grid. One of these cases was an extreme event of magnitude 9.0 with epicenter in the Makran Subduction Zone (MSZ). For the capital city area, Muscat, the resultant flooding map is shown in Fig. 19; the profiles that were selected for the validation are superimposed on this map. The tsunami wave height and period offshore were approximately 2 m and
2300 s. In these cases, the measured run-ups at each profile were 6.2, 8.7, and 7.7 m. The corresponding run-ups calculated with the new database were 6.3, 8.5, and 7.8 m.


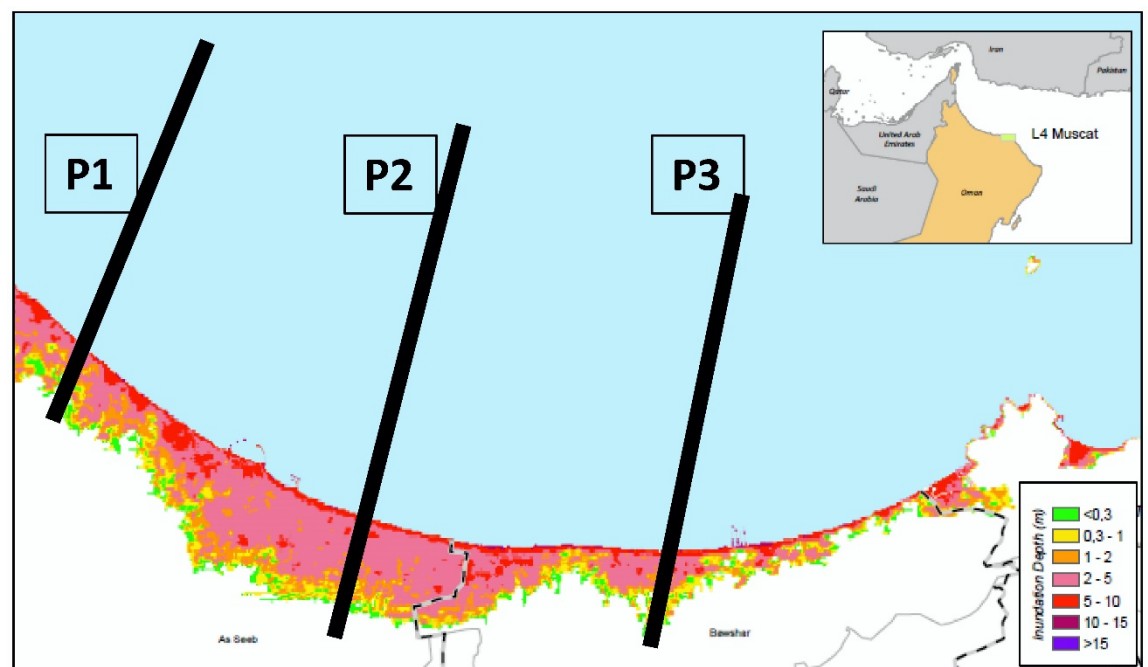

**Fig. 19. Flooded area in the municipality of Muscat, capital city of Oman, due to a 9.0 magnitude event with epicenter in the Makran Subduction zone. The extracted locations where run-up was estimated and the run-up values, both modeled and estimated with the IH-TRUST tool, are provided in Table 4.**

5    Table 4 summarizes the results obtained for the validation with the high-resolution simulations in the three scenarios. The run-up values, both those calculated with the numerical model and those estimated with the proposed database and detailed methodology described above, have a similar magnitude; in some cases, the result is accurate enough to rely on the results of the presented methodology.

**Table 4. Tsunamis scenarios included in the validation process of the database and tool. The numerical model column includes the run-up obtained with the high-resolution numerical simulations and can be compared to the estimations from the application of the IH-TRUST and Synolakis formula**

| PLACE | PROFILE | Coordinates of the run-up point | | Run-up (m) | | |
|---|---|---|---|---|---|---|
| | | LON | LAT | NUMERICAL MODEL | IH-TRUST | SYNOLAKIS |
| **Trujillo (Perú)** | P1 | -79.076 | -8.114 | 10.6 | 10.5 | 8.2 |
| | P2 | -79.037 | -8.134 | 8.9 | 8.8 | 8.1 |
| | P3 | -79.000 | -8.166 | 12.8 | 11.6 | 9.9 |
| | P1 | 58.211 | 23.657 | 5.3 | 6.2 | 6.9 |
| | P2 | 58.269 | 23.600 | 5.5 | 6.1 | 7.0 |





| | | | | | | |
|---|---|---|---|---|---|---|
| **La Libertad (El Salvador)** | P3 | 58.389 | 23.597 | 6.3 | 7.0 | 6.9 |
| **Muscat (Oman)** | P1 | -89.329 | 13.483 | 6.2 | 6.4 | 8.2 |
| | P2 | -89.318 | 13.487 | 8.7 | 8.6 | 8.8 |
| | P3 | -89.283 | 13.486 | 7.7 | 7.9 | 10.4 |

Fig. 20 shows this comparison in a plot, in which the fitting between the modeled and calculated run-up values is noted. In addition, the estimated run-up from the new methodology is better than the result of the Synolakis formula, which generally overestimates the run-up.



**Fig. 20. Plot of the numerically modeled run-up against the calculated run-up values for the validation cases.**

## 5.2 Validation with data recorded during field campaigns after real events

The low frequency of major tsunamis are invaluable to the field campaigns that are carried out immediately after a tsunami

5   event occurs. These field campaigns allow the evaluation of the developed strategies of risk reduction and the creation of new, more accurate strategies. From a pragmatic point of view, the data collected during these campaigns allow scientists and engineers to validate or calibrate numerical models and methodologies. In this case, this type of validation has been addressed



using the available field data of the events in Japan (2011) and Chile (2010 and 1960). The bathymetric profiles used in the validation have been constructed using GEBCO. The tsunami wave time series have been obtained from the data available from DART buoys (Meinig et al., 2005) or numerical simulations of accurate sources; this process is explained in detail later in the paper. The results of the application of the methodology have been compared to observational data recordings and field

survey papers.

### 5.2.1   2011 tsunami on the coast of Japan affecting the Pacific basin

On the 11th of March, 2011, a 9.0 earthquake, which had an epicenter close to the coast of Japan, triggered a tsunami that reached the coast of Japan within one hour. This tsunami wave propagated across the Pacific Ocean, reaching the U.S. West Coast in 10 hours and the coast of Chile in 21 hours.

The tsunami wave time series used for this validation have been obtained from the data available from DART buoys (Meinig et al., 2005). The results were compared with the observed run-up (National Geophysical Data Center NOAA).

It is essential to highlight that the application of the new run-up estimation methodology is restricted to the profiles and wave shapes whose parameters fall inside the ranges covered by the database (see Table 1). Therefore, the use of the methodology

is limited to these cases. An example of non-applicability occurs when the tsunami height and period are obtained ($d_2$) in a shallow area of the ocean or when the generation zone is too close to the study area and a complete time series of the tsunami wave cannot be properly recorded at an adequate depth.



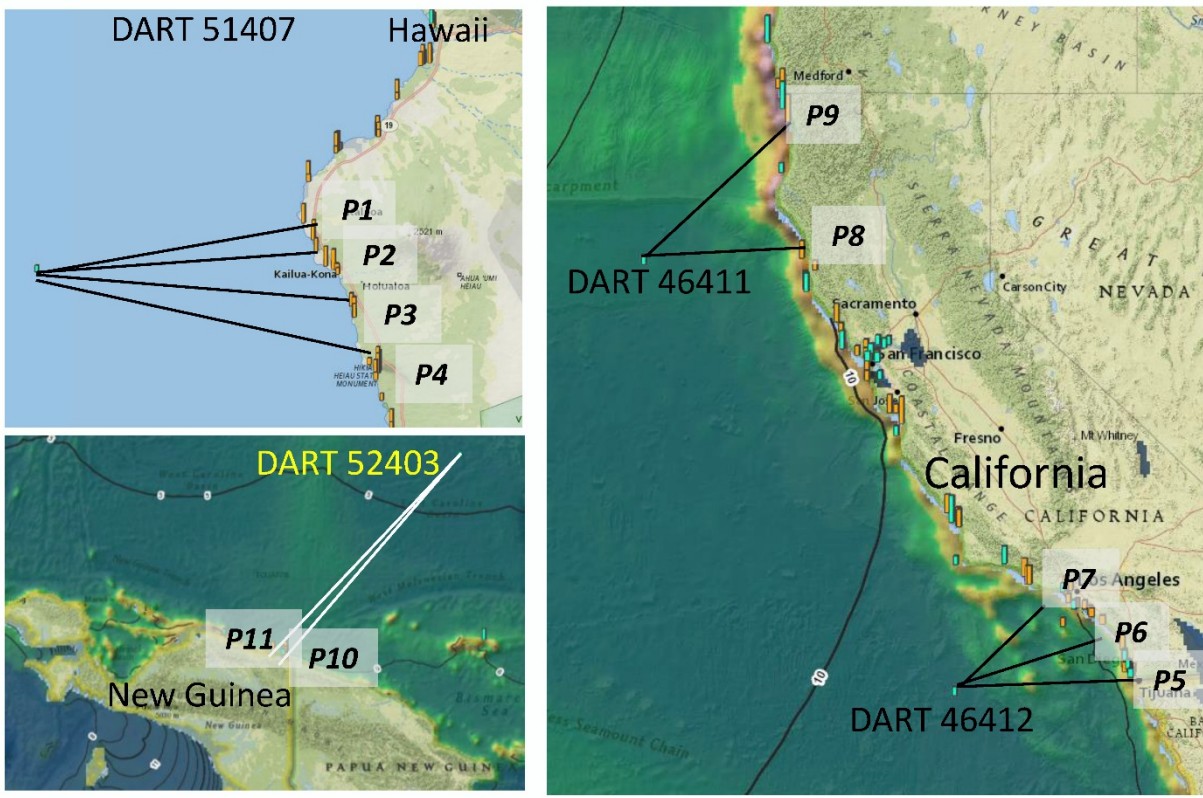

**Fig. 21. Validation with DART buoy time series. 4 DART buoys were used, and their data were applied to several bathymetric profiles to validate the methodology. The locations of the points where the run-up was estimated are included in Table 5**

In the case of the Japan 2011 event, due to the proximity of the coast, it was not possible to obtain a complete time series between the epicenter and Japan, and the validation has been carried out in other areas of the Pacific Ocean, using four DART buoy records (near Hawaii, California, and Papua-New Guinea). The names and locations of the DART buoys used are given in Table 5. This table also includes the names and locations where the run-up was estimated with the data of each DART buoy, the run-up value recorded in the field surveys at those locations, and the estimated value of the run-up, both by using the new

methodology and by applying the Synolakis formula (calculating the tsunami wave height at a depth of 10 m using Green's law. The buoy locations are also included in Fig. 21.

**Table 5. Validation with DART buoy time series of the Japan 2011 event. 4 DART buoy datasets were used on several bathymetric profiles to validate the methodology. Location names correspond to those given by the National Geophysical Data Center**

**(NOAA). Synolakis run-up was estimated by applying the so-called Green's Law to the time series of the DART buoys to obtain the tsunami height near the coast.**





| DART Buoy | | | | | Run-up (m) | | | | |
| # | LON | LAT | DEPTH (m) | LOCATION | LON | LAT | SURVEY | Synolakis + Green | IH-TRUST |
|---|---|---|---|---|---|---|---|---|---|
| **51407** | -156.5 | 19.620 | 4771 | **P1 Wawalolo** | -156.05 | 19.71 | 2.4 | 3.8 | 2.0 |
| | | | | **P2 OldAirport** | -156.01 | 19.64 | 3.1 | 3.8 | 2.0 |
| **Hawaii** | | | | **P3 Kahaloo** | -155.97 | 19.58 | 2.0 | 3.8 | 2.4 |
| | | | | **P4 Keel** | -155.93 | 19.46 | 3.0 | 3.8 | 2.8 |
| **46412** | -120.7 | 32.250 | 3776 | **P6 Ocean Beach** | -117.26 | 32.74 | 1.0 | 1.9 | 1.5 |
| | | | | **P7 Marina del Rey** | -118.45 | 33.98 | 1.0 | 1.9 | 1.9 |
| **California** | | | | **P5 Channel Islands** | -119.22 | 34.15 | 1.2 | 1.9 | 1.4 |
| **46411** | -127.0 | 39.340 | 4319 | **P9 Jenner River** | -123.1 | 38.43 | 1.0 | 2.7 | 1.1 |
| **California** | | | | **P8 Dolphin isle** | -123.8 | 39.43 | 1.0 | 2.7 | 1.3 |
| **52403** | 145.52 | 4.020 | 4474 | **P10 Holtekamp** | 140.779 | -2.627 | 2.0 | 2.3 | 1.7 |
| **Papua-New Guinea** | | | | **P11 Pelabuham** | 140.368 | -2.461 | 1.3 | 2.3 | 1.4 |

As it can be inferred from the application of the methodology, the run-up estimated values are on the same order of magnitude as the recorded inundation; generally, the results are accurate. These results are also closer to the observed run-ups than those obtained by applying Synolakis formula, which often overestimates the run-up.

### 5.2.2 Chilean coast tsunamis (2010 and 1960)

When no real record is available to determine the offshore wave shape (DART buoys), the main issue is the correct definition of the source to compute an accurate numerical simulation. Although there is no shortage of uncertainties in the determination of the source, the tsunami initial surface deformation models that have been developed are accurate (Barrientos and Ward, 1990). As an alternative validation approach, two of these models have been used to validate the new run-up estimation methodology with the events that occurred in Chile in 2010 and 1960.





**Fig. 22.** **Profiles and locations used in the validation of the new methodology by using the run-up recorded after the tsunami events of 2010 (a) and 1960 (b) in Chile. The locations of the points where the run-up was estimated are included in Table 6**

5  On the 27th of February, 2010, an 8.8 magnitude earthquake with epicenter on the coast of Chile triggered a tsunami that reached the Chilean coast in less than 30 minutes. In the Bio-Bio region, the run-up was recorded at several locations (Fritz et al., 2011). To apply this methodology, first, a rough numerical simulation of the generated tsunami was addressed. This simulation used the source definition by (Shao et al., 2010) and gridded the GEBCO data with 700 m cells (see Fig. 22). From this simulation, the profiles and wave amplitude time series in the generation area were obtained. The tsunami wave height



and period recorded at each location and the result of the interpolation from the further improved database for each corresponding profile are given in Table 6.

The optimal application of the run-up estimation methodology is achieved at the locations sufficiently far from the source, as explained in the previous subsection. The result at these points (1, 2, 3, 8, 9 and 10) have the same order of magnitude of the recorded run-up from Fritz's survey. In the locations in front of the source, the initial deformation of the water surface did not allow a complete time series to be obtained to estimate the tsunami wave height and run-up.

Regarding the 1960 earthquake and tsunami in Chile (Lomnitz, 2004) (Fig. 22), this earthquake is considered the greatest earthquake ever recorded, and the numerical simulation computed for the validation used the source by Barrientos and Ward (1990). The run-up data for the validation was obtained from the NOAA global historic tsunami database. In this case, the data are mainly from eyewitness testimony.

In Table 6, the results of the application of the methodology at 7 locations and the recorded run-up are given. In this case, the tsunami wave height at 3 of the locations (P13, P14, and P15), was such that the profiles were not within the database application ranges. The other 4 locations provided results that are on the same order of magnitude as the observed run-up.

In the application of the new methodology to the Chile events, the tsunami wave height used for the interpolation came from a numerical simulation, and the results were compared to real run-up records. Although the validation inherits the uncertainties of the source, the results are sufficiently accurate, taking into account the limitations explained above.

**Table 6. Validation of the methodology with the results of numerical simulations of realistic sources of the 1960 and 2010events on the coast of Chile.** Fritz et al. (2011) **survey results were used to validate the results from the new methodology for the Chile 2010 event. NOAA´s National Geophysical Data Center data were used to carry out the comparison with the 1960 event**

|  | PROFILE | LOCATION | LON | LAT | Run-up (m) | |
|---|---|---|---|---|---|---|
|  |  |  |  |  | SURVEY | IH-TRUST |
|  | P1 | Ritoque | -71.528 | -32.826 | 3.4 | 1.39 |
|  | P2 | Cartagena | -71.602 | -33.542 | 4 | 1.93 |
|  | P3 | El Yali | -71.717 | -33.751 | 2.1 | 3.47 |
|  | P4 | Pichilemo | -72.005 | -34.384 | 5 | N/A |
|  | P5 | Llanco | -72.623 | -35.584 | 11.4 | N/A |
| **2010 event** | P6 | Mela | -72.852 | -36.36 | 3.1 | N/A |
|  | P7 | Playa | -72.911 | -36.478 | 6.6 | N/A |
|  | P8 | Ranquil Bajo | -73.596 | -37.526 | 5.7 | 2.2 |
|  | P9 | Mouth of Lieu Lieu | -73.449 | -38.097 | 2.3 | 2.2 |



| | | | | | |
|---|---|---|---|---|---|
| | P10 | **Puerto Saavedra** | -73.701 | -38.783 | 2.5 | 2.3 |
| | P11 | **Tome** | -72.962 | -36.619 | 2.5 | 3.4 |
| | P12 | **Lebu** | -73.674 | -37.608 | 4 | 4.6 |
| | P13 | **Punta Saavedra** | -73.407 | -37.608 | 11.5 | N/A |
| **1960 event** | P14 | **Valdivia** | -73.411 | -39.844 | 10 | N/A |
| | P15 | **Ancud** | -73.828 | -41.859 | 12 | N/A |
| | P16 | **Chiloe** | -74.176 | -42.465 | 10 | 9 |
| | P17 | **Guafo** | -74.83 | -43.578 | 10 | 10.2 |

### 6. Conclusions

The calculation of the flooding that a tsunami causes inland is addressed when a tsunami risk assessment is conducted. For a historical event, the assessment determines the limit of the affected area. In addition, the predictive evaluation of this flooded area, based on the potential tsunami scenarios that can affect it, allows prevention and mitigation measures to be established, helping to reduce the risk.

However, the calculation of this flooded area, particularly the assessment of the run-up, is not always direct. Occasionally, there are no high-resolution data that allow the application of numerical models. In addition, the accuracy of the existing empirical formulae can be improved, since they do not take into account natural topobathymetric profiles from the propagation to the inundation areas.

In this paper, an alternative methodology that complements the existing ones has been presented. This methodology consists of a numerical flume formed by the coupling of two numerical models (COMCOT and IH2VOF). The developed hybrid model is applied to each part of the generation-propagation-inundation process and this numerical model obtains a more accurate result; additionally, it is computationally affordable. The inputs for this hybrid model are the topobathymetric profile and the tsunami wave. The topobathymetric profiles were parameterized with 5 parameters (3 slopes and 2 depths), using a real profile sample to define the parameterization. In addition, the tsunami waves were parameterized with 2 parameters (tsunami height and period) using tsunami amplitude time series obtained by using numerical simulations of realistic tsunami events.

This methodology allows the accurate calculation of the run-up on along topobathymetric profile. Therefore, this methodology has been used to construct a tsunami run-up database. This database aimed to create an interpolation domain in which new run-up calculations could be carried out. The events of the database are combinations of a selection of bathymetric profiles



and tsunami waves that were simulated with the hybrid model to create the database of simulations from which an interpolation can be executed to calculate the run-up of new tsunami scenarios.

To easily address the interpolation, a tool called IH-TRUST was scripted. This tool uses real profile and wave data,

5    parameterizes them to find their most similar parameters in the database, and interpolates the results to provide a run-up value.

To validate this new methodology and tool, the results of its application have been compared with both high-resolution numerical simulations and field survey data. The run-ups obtained with IH-TRUST are consistent and suggest that the tool can accurately calculate the run-up.

The assessment of the tsunami hazard begins by calculating the area affected by the tsunami. In those coastal areas where no other data are available, the detailed methodology and tool allow the run-up value of tsunami events to be determined without using high-resolution numerical simulations.

Therefore, to assess the hazard in a tsunami-prone area, this methodology can be applied to several profiles along the coastal area study. As a result, the methodology provides the run-up at each of the profiles, allowing an estimation of the flooded area from an area within the envelope of the run-up limits.

20    The application of the tool has some limitations; for example, the tool will indicate if the bathymetric profile or the tsunami wave parameters are not included within the range of values in the database.

New work in this field should take into account these difficulties to further develop the database with new parameter values that include these singularities.

The generation of the database and the values of run-up obtained from a combination of the bathymetric profiles and tsunami waves have provided a rich and populated space where the influence of each parameter on the final value of the run-up can be addressed. In this sense, which profiles are more prone to suffer higher run-ups in the case of a tsunami can be defined. For instance, profiles with high land slopes ($tan\beta_0$) are associated with higher run-up values than those with low land slopes. In

30    addition, some combinations of offshore slopes and continental shelf slopes ($tan\beta_1$ and $tan\beta_2$) minimize the run-up value for the same tsunami wave. In addition, the influence of $tan\beta_2$ is considerable and justifies inclusion of the deep-water area ($d_2$) in the parameterized profile. On the other hand, when the profile is for a large continental shelf, the run-up increases; however, the run-up value decreases for gentle continental shelf slopes.



Traditionally, empirical methods, like the application of Synolakis´s formula, simplify the profile using one or two slopes (Park et al., 2015). However, this assumption is not accurate; in this study, the importance of using a complete profile, including the tsunami generation area, has been noted, as well as the influence of the profile parameters on the final run-up value.

5 **Acknowledgements**

The research leading to these results has received funding from the European Union's Seventh Framework Programme (FP7/2007-2013) under grant agreement n° 603839 (Project ASTARTE - Assessment, Strategy and Risk Reduction for Tsunamis in Europe).

**Appendix A**

**Database profiles**

15    In this section, the 49 artificially generated profiles are shown in Figure A1. The five corresponding parameters are listed on Table A1.







**Fig. A1. The 49 profiles used to generate the IH-TRUST database**

**Table A1. Synthetic profiles**

| # | $\tan\beta_0$ | $\tan\beta_1$ | $\tan\beta_2$ | $d_1$ | $d_2$ |
|---|---|---|---|---|---|
| 1 | 15.00% | 2.50% | 2.50% | 0 | 2600 |
| 2 | 0.05% | 20.00% | 20.00% | 0 | 3000 |
| 3 | 0.05% | 1.00% | 8.50% | 1100 | 4800 |
| 4 | 8.00% | 10.00% | 10.00% | 0 | 6000 |
| 5 | 0.50% | 0.15% | 12.00% | 150 | 2600 |
| 6 | 0.05% | 0.15% | 2.50% | 200 | 5200 |
| 7 | 10.00% | 0.50% | 14.00% | 500 | 4300 |
| 8 | 10.00% | 14.00% | 14.00% | 0 | 2600 |
| 9 | 15.00% | 0.15% | 4.00% | 50 | 5200 |
| 10 | 0.05% | 10.00% | 10.00% | 0 | 4300 |
| 11 | 5.00% | 1.00% | 1.00% | 0 | 3000 |
| 12 | 10.00% | 0.50% | 14.00% | 20 | 2600 |
| 13 | 8.00% | 1.50% | 4.00% | 800 | 5200 |
| 14 | 15.00% | 10.00% | 10.00% | 0 | 4300 |
| 15 | 0.05% | 2.00% | 16.00% | 500 | 4300 |
| 16 | 5.00% | 0.05% | 12.00% | 50 | 5200 |
| 17 | 10.00% | 0.50% | 2.50% | 500 | 3500 |
| 18 | 0.50% | 1.50% | 4.00% | 500 | 3500 |
| 19 | 8.00% | 1.00% | 2.50% | 200 | 6000 |
| 20 | 5.00% | 16.00% | 16.00% | 0 | 4300 |





| | | | | | |
|---|---|---|---|---|---|
| 21 | 3.00% | 10.00% | 10.00% | 0 | 2600 |
| 22 | 8.00% | 7.00% | 7.00% | 0 | 3900 |
| 23 | 1.00% | 7.00% | 7.00% | 0 | 6000 |
| 24 | 5.00% | 1.00% | 10.00% | 800 | 3500 |
| 25 | 5.00% | 0.50% | 18.00% | 200 | 3500 |
| 26 | 1.00% | 1.50% | 10.00% | 500 | 5600 |
| 27 | 12.00% | 1.00% | 12.00% | 200 | 5600 |
| 28 | 15.00% | 0.30% | 10.00% | 20 | 3900 |
| 29 | 15.00% | 8.50% | 8.50% | 0 | 6000 |
| 30 | 1.00% | 1.00% | 7.00% | 20 | 3900 |
| 31 | 0.05% | 2.50% | 20.00% | 20 | 2600 |
| 32 | 0.05% | 0.05% | 16.00% | 20 | 4300 |
| 33 | 15.00% | 10.00% | 10.00% | 0 | 2600 |
| 34 | 8.00% | 0.15% | 2.50% | 100 | 4300 |
| 35 | 5.00% | 1.50% | 4.00% | 1100 | 3900 |
| 36 | 5.00% | 1.50% | 7.00% | 300 | 2600 |
| 37 | 10.00% | 1.50% | 10.00% | 500 | 3000 |
| 38 | 8.00% | 1.50% | 12.00% | 800 | 5200 |
| 39 | 8.00% | 0.05% | 12.00% | 20 | 3900 |
| 40 | 0.05% | 4.00% | 4.00% | 0 | 2600 |
| 41 | 2.00% | 0.50% | 4.00% | 800 | 5200 |
| 42 | 3.00% | 2.00% | 14.00% | 500 | 3000 |
| 43 | 10.00% | 7.00% | 7.00% | 0 | 2600 |
| 44 | 5.00% | 2.00% | 8.50% | 300 | 4300 |
| 45 | 5.00% | 4.00% | 4.00% | 0 | 5200 |
| 46 | 0.05% | 0.05% | 10.00% | 100 | 6000 |
| 47 | 10.00% | 14.00% | 14.00% | 0 | 4800 |
| 48 | 15.00% | 0.30% | 2.50% | 20 | 3900 |
| 49 | 10.00% | 0.50% | 1.00% | 200 | 2600 |

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
