# Peer review of "Tsunami run-up estimation based on a hybrid numerical flume and a parameterization of real topobathymetric profiles"

_Natural Hazards and Earth System Sciences, 2017_

## Referee Comment (RC1) · Anonymous Referee #1 · 1 Jan 2018

The manuscript proposes a methodology to estimate tsunami runup by mixing up a classical Tsunami code (COMCOT), for the first stages, and an averaged Navier-Stokes model for the runup process. In my opinion, the manuscript exhibits a well and organized work and I suggest that it should be published after minor revision regarding specific points that shold be clarified, because they affect in the understaing and make the manuscript not fully reproducible.

Major comments:

1) Time computation is regularly mentioned, however there is no solid numbers. For example, how long it takes a regular tsunami running? How long it takes obtain the

final runup estimation with the presented methodology ?

2) It is also not mentioned, but I guess authors have assumed an instantaneous tsunami generation. This have to be very clear. In general, there is lack of details on the tsunami modeling. Domain size, computation time, CFL condition (depending on your chosen grid size), etc. You should, at least, comment some lines due to the fact that time characteristics of the seismic source can enhance the tsunami amplification. This becomes important in huge and rare events as The 1960 Chilean Earthquake and 2004 Sumatra Earthquake, where the source time function is not well resolved (specialy in the Chilean event). Besides of all the earthquake parameters, there is the slip distribution. It is demostrated that the runup can be amplified up to six times (Geist (2002), Ruiz et al. (2015)). So, the kind of seismic sources should be clearly defined.

3) The approximation of the topo-bathymetryc profiles are fitted from the GEBCO data, but no resolution is mentioned. The authors fixed the profiles with four (4) segments: a constant depth (1) conected to two lines offshore (2) and another line inland (1). The first and natural question is why to set 4 segments ?? Is it because the 5-space of parameters is already big enough? Another issue, is that the trench morphology is not captured, or at least, not showed in the manuscript. This is because in sibduction zones, before the ocean becomes "constant", there is a huge depression, especially in The Marianas trench, where the water column is higher and faster.

4) Authors "cheats" the tsunami interaction of the reflected wave by assuming a constant and flat region with open boundary condition. However, would not this add some kind of artifacts to the model? Test regarding this issue should be do it.

5) Authors make use of analytical solution of Synolakis (1987), however, I'm not convinced that is the good one here. There are analytical solutions in piecewise bathymetries (e.g. Kanoglu & Synolakis (1998), Fuentes et al. (2015), Riquelme et al. (2015)). Actually, in figure (13) the results do not agree with those analytical solution which state that offshore slope closest to the coast controls the runup process.

6) It is not mentioned the criterion to trace the profiles. Perpendicular to the shore? Paralel to the wave travel??

7) The methodology is compared with numerical models and retrieves same estimations. The fact that inacessible high-resolution data can be overcame should be more highlighted. Again, I dont think Synolakis's formula is comparable here, since it uses a Solitary wave as initial condition, and there are analytical solutions that can handle with arbitrary shapes (Madsen & schaffer,(2010) , Fuentes (2017) ).

Specific comments:

- First line of intro: Add the 2010 Chile tsunami.

- Page 9, 5: It seems that "H" is unnecessary here. Also, it should be clarified that period relation is valid in the linear regime.

- Please add geographic axis to the map plots.

- Page 36, 5: Authors say "the results are accurate". Please, add a percentage based on the results.

References:

- Kanoglu, U. & Synolakis, C.E., (1998). Long wave runup on piecewise linear topographies, J. Fluid Mech. (JFM), 374, 1–28.

- Fuentes, M., J. Ruiz, and S. Riquelme (2015), The run-up on a multilinear sloping beach model, Geophys. J. Int., 201(2), 915–928.

- Fuentes M., (2017). Simple estimation of linear 1+1 D long wave run-up. Geophys. J. Int., 209(2), 597-605.

- Geist, E. (2002), Complex earthquake rupture and local tsunamis, J. Geophys. Res., 107(B5). - Riquelme S., Fuentes M., and Hayes G., (2015). A rapid estimation of near-field tsunami runup. Jouranl of Geophysical Research: Solid Earth, 120(9), 6487-6500.

- Ruiz, J., M. Fuentes, S. Riquelme, J. Campos, and A. Cisternas (2015), Numerical simulation of tsunami runup in northern Chile based on non-uniform k−2 slip distribution, Nat. Hazards,1–22.

———————————————

---

## Referee Comment (RC2) · Anonymous Referee #2 · 29 Jan 2018

General comments: This paper presents a method for quickly assessing tsunami run-up for different tsunami wave shapes and bathymetries. The method is build on a hybrid approach, where the Non-Linear Shallow water model COMCOT is used for the deep-water propagation and a RANS models is used for the near shore processes. The results from this hybrid model, then enters an interpolation model, which can be used to assess run-up. The approach is novel and innovative, and I especially like adding a fast interpolation model. I have however, a few major concerns. The actual implementation of the hybrid model is not well described and I think the coupling between the two models can pose big problems. Further, the hybrid model is not validated on its own in a controlled environment.

[Figure]

Major comments:

1) Parts of implementation and usage of the hybrid model is not well described.

a. What are typical grid sizes in the RANS model? Are these sufficient to handle the processes, which NLSW models cannot handle? Like wave breaking.

b. What are typical $L_x$ lengths?

c. How are the boundary conditions for the turbulence mode?

d. It is unclear how $x_{cut}$ is determined. In the paper two criteria is given. One is to maximize the area of the IH2VOF domain and the other is to ensure that flooding does not exceed the inland end. Regarding the first criteria, letting IH2VOF cover the entire numerical flume would achieve that, but that is clearly not what is being done. Regarding the latter, I cannot see how the end position of the IH2VOF domain influences the position of $x_{cut}$.

e. One of the advantages of the hybrid model is that the RANS model can handle processes that the simpler COMCOT model cannot. One of such processes is the wave splitting into an undular bore, which can happen when the wave travels long in shallow water and this has been witnessed in many real life tsunamis. To be able to capture this effect $x_{cut}$ needs to be positioned sufficiently off shore. How is this ensured?

f. To avoid reflection, the numerical flume of COMCOT is altered, to properly access the incoming wave. I have a problem with this approach. In reality, especially in cases with steep slopes, there will be significant reflection from the beach which will and should affect the incoming wave. This effect cannot be captured with the current approach. Further it is also unclear what would happen when the reflected wave from the IH2VOF domain meets the hard boundary between the two models. Will this cause additional reflections in IH2VOF domain?

g. The calculations of L does not match Fig. 4. E.g. $L_i$ is given as $L_i=1/50 \tan(\beta_0)$.

This will result in a very low $L_i$. Further $L_{off}$ is given as $L_f + x_2$, but according to Fig. 4 it should be $L_{off} = L_f + x_2 + x_1$. Finally, there is no need for $\delta X$ in the calculations of $L_f$ as it is present both in the denominator and the numerator.

h. How is the run-up height determined in the IH2VOF model? In a VOF computation, the interface can span across several cells.

2) The first validation case is performed by comparing the interpolation model to the hybrid model. This is an important and satisfying validation case, but I am lacking validation of the actual hybrid model. How will the model perform using the approach outlined in the paper e.g. in cases with both breaking and non-breaking waves running up a constant slope.

3) The performance of the iterative solver is compared to the Synolakis formula. I do not believe this is a fair comparison, as it is created for the run-up of a solitary wave, which as highlighted by the author does represent a real geophysical tsunami event. A more relevant comparison could be to the analytical model proposed by Madsen and Schäeffer (2010), as also highlighted by the authors in the introduction. (The Synolakis formula require a proper reference).

4) The periods are estimated as the time between the first two zero crossings for positive heights. Does this mean that the model cannot differentiate between tsunamis having only positive surface displacement and e.g. a leading depression?

5) With this approach of estimating period and wave height, I see a potential problem in the case where the leading wave is not the largest. Can you please elaborate on this?

6) One of the main points off this work is to be able to quickly access tsunami run-up without doing long complicated simulations. Therefore for this work to fulfill this, it would be beneficial is the TRD database was made available to engineers. Are there any plans regarding this?

Smaller comments:

1) Page 1, line 8: It is stated that Run-up is accurately calculated by means of numerical models. This is a rather strong statement. I would prefer it rewritten as: can be accurately calculated

2) Page 1, line 14. The models here, and several other places are referred to as schemes. They are however not schemes, but models. Please change the formulations.

3) Fig. 5. It is stated that only the COMCOT model is used with the altered domain. If this is indeed the case, then please remove the IH2VOF domain from the figure, as it is causing confusion.

4) Fig. 6 units and legends are missing on the colorbars. Please add these.

5) Page 14, line 2. It is stated that $d_2 - d_1$ was always shorter than 2200 m and $x_1$ shorter than 210 km. How does this correspond with table A1 where $d_2 - d_1$ is always larger than 2200 m?

6) Fig 8. Many of the axis are missing units. Please add these.

7) Fig 10. Please add missing units to the axis.

8) Page 19, line 6. It is stated that T corresponds to the time between the first two zero crossings for positive heights. However in Fig 11. It looks as if the second zero crossing has not occurred within the shaded area?

9) Page 24. It is described how low values of $\tan(\beta_2)$ gives lower run-up height due to friction. From Fig 13, it can however also be seen that the run-up heights reduce with large values of $\tan(\beta_2)$. Please elaborate on this.

10) It is unclear exactly what Fig 15 is describing. Please rewrite the description for clarity and add units to the axis.

---

## Author Comment (AC1) · 13 Feb 2018

Anonymous Referee #1 The manuscript proposes a methodology to estimate tsunami runup by mixing up a classical Tsunami code (COMCOT), for the first stages, and an averaged NavierStokes model for the runup process. In my opinion, the manuscript exhibits a well and organized work and I suggest that it should be published after minor revision regarding specific points that shold be clarified, because they affect in the understaing and make the manuscript not fully reproducible.

REPLY: We thank reviewer 1 for the thorough review of our manuscript and the positive comments regarding its organization. We have added their ideas and modifications to

the paper (green color in the attached revised version), increasing its overall significance.

Major comments:

1) Time computation is regularly mentioned, however there is no solid numbers. For example, how long it takes a regular tsunami running? How long it takes obtain the Final run-up estimation with the presented methodology?

REPLY: Thanks a lot for coming up this topic. Computational time information is basic to have a more global approach of the process of construction of the database, and to understand how it makes easier and faster the run-up calculation. On hazard assessments, in particular of large areas, the computational time become a key element on the methodology to apply. For example, in the case of a simulation of an event that travels through the ocean basin and then floods a local area, it can require several levels of nested grids to simulate the tsunami including a high resolution grid for the local area. The computational time to conduct this simulation depends on many aspects but it can take 10 to 16 hours in a common computer. In the methodology presented in this paper, the interpolation itself takes just some seconds. If a numerical simulation with SWE model is carried out to obtain the wave conditions to use them as input for the IH-TRUST, then just a single grid for the whole ocean basin is necessary, what could take around one hour, depending again on the simulation domain. Finally, the simulations that are of the database required a long time for calculation depending mainly on the size of the VOF domain. Typical times range from 2 hours to 16 hours. This data has been added to the new version of the manuscript in the conclusions section (page 41, lines 6-8)

2) It is also not mentioned, but I guess authors have assumed an instantaneous tsunami generation. This have to be very clear. In general, there is lack of details on the tsunami modeling. Domain size, computation time, CFL condition (depending on your chosen grid size), etc. You should, at least, comment some lines due to the fact

that time characteristics of the seismic source can enhance the tsunami amplification. This becomes important in huge and rare events as The 1960 Chilean Earthquake and 2004 Sumatra Earthquake, where the source time function is not well resolved (specially in the Chilean event). Besides of all the earthquake parameters, there is the slip distribution. It is demostrated that the runup can be amplified up to six times (Geist (2002), Ruiz et al. (2015)). So, the kind of seismic sources should be clearly defined.

REPLY: As the reviewer correctly addresses, the tsunami generation follows some hypothesis or simplifications. Specifically, an instantaneous generation and a regular and constant slip distribution were assumed. In this way, it is interesting to highlight, as the reviewer remarks, that when historic or past events are being simulated a proper source could be evaluated and used to determine the H and T of the tsunami wave to be used as an accurate input for the methodology. On the other hand, potential events that are part of tsunami hazard assessments commonly use idealized sources, that can be used to evaluate H and T. Anyway, the tsunamigenic sources used for elaborating the database were idealized parameterizations that were transformed into initial water surface displacement by means of Okada model. Regarding the grid size for COMCOT simulations, it was set to $\triangle x=500m$. Depending on the maximum depth of the grid the necessary time step to satisfy Courant Condition was calculated and used, based on the restriction of the model for the condition Cr=0.5: $(C·\triangle t)/\triangle x<C\_r$,where c=$\sqrt{(g·h)}$ Clarifying lines have been added to the manuscript (page 9 and page 14)

3) The approximation of the topo-bathymetryc profiles are fitted from the GEBCO data, but no resolution is mentioned. The authors fixed the profiles with four (4) segments: a constant depth (1) conected to two lines offshore (2) and another line inland (1). The first and natural question is why to set 4 segments?? Is it because the 5-space of parameters is already big enough? Another issue, is that the trench morphology is not captured, or at least, not showed in the manuscript. This is because in subduction zones, before the ocean becomes "constant", there is a huge depression, especially in The Marianas trench, where the water column is higher and faster.

REPLY: Regarding the GEBCO bathymetry, the topobathymetric profiles that were used in the elaboration of the database were obtained from GEBCO, using the resolution provided by this database, 30" (around 900 m in the Equator). Regarding the chosen geometry, we analyzed worldwide profiles trying to find a parameterization that covered two main aspects. First, and mainly, that they could represent appropriately most of the profiles and second, that the selected parameterization allowed managing the database to be created. The technique for classification (Maximum dissimilitude) and interpolation (radial basis functions) are specially designed to work on high dimensional domains (i.e. Camus et al., 2011), therefore the number of segments is not an issue. Nevertheless, a run-up calculation requires the parameterization of a profile as input, therefore that parametrization must be functional. After considering other options, like adding a new segment, we considered that our parameterization achieved this equilibrium between representability and functionality. Regarding the subduction trench, its applicability, as it can be seen in the applications cases is limited to those generation areas that are deep enough to be part of a profile included in the database. In this sense, the system works quite well if this is the case, as it was observed in the examples of Chile. However, there is a limitation, well noted by the reviewer: the profile parameterization falls out of the application range (see page 34 line 13). The result of assuming that the seabed is constant seawards the generation area gives a good approximation of the "trench problem".

Some lines explaining these aspects have been included in the new version of the manuscript (page 7, 5, 35.) and in the conclusions section

4) Authors "cheats" the tsunami interaction of the reflected wave by assuming a constant and flat region with open boundary condition. However, would not this add some kind of artifacts to the model? Test regarding this issue should be do it.

REPLY: One of the biggest issues to perform the coupling between models was to obtain a clean input wave for forcing the VOF domain. The problem arises because the wavelength of tsunami waves is, in some cases, longer than VOF model domain. This

implies that before a tsunami wave passes completely through the boundary between models, the wavefront reaches the coast, is reflected and return to the initial boundary aliasing the wave amplitude. Several tests have been run to assure that this artifice of assuming plane beach and open boundary do not affect the numerical simulation. This artifice avoids the interaction of the tsunami wave arriving to the coast and the reflected wave, "cleaning" optimally the signal of the tsunami waves that were included in the development of the database, as explained in figure 6.

5) Authors make use of analytical solution of Synolakis (1987), however, I'm not convinced that is the good one here. There are analytical solutions in piecewise bathymetries (e.g. Kanoglu & Synolakis (1998), Fuentes et al. (2015), Riquelme et al. (2015)). Actually, in figure (13) the results do not agree with those analytical solution which state that offshore slope closest to the coast controls the runup process.

REPLY: We really appreciate this comment. We have used Synolakis as an example of comparison because although it was created for Solitary waves it has been commonly used on tsunami risk assessments, despite its application is not appropriate as highlighted by Madsen (2008). In order to improve this, we have added a new column to the comparison tables with the value of the run-up but calculated with Madsen and Shaffer (2010) for single waves, as also suggested by Reviewer 2. In addition, we have included the given references in order to have a complete view of the existing solutions, In the case of solitary waves, it was shown by Kanoglu and Synolakis(1998) that the slope closest to the coast controls the run-up. Our results show that, actually, the slope closest to the coast is very important in the final value of the tsunami run-up. However, according to our results, in the case of tsunami waves the influence of the other slopes, especially the one of the next segment seawards should not be neglected, remarking again the difference in the wavelength of solitary waves and tsunamis, what leads to a different behavior, a different time while the wave is affected by each segment of the bathymetry, affecting the reflection, shoaling etc. This aspect, also noted in Naik &Behera (2016) using numerical models, has been also added to the manuscript, in page

26 and in the discussion section.

6) It is not mentioned the criterion to trace the profiles. Perpendicular to the shore? Paralel to the wave travel?? REPLY: Orientation of profile is a key piece on the runup calculation using this method. As the database was constructed using a numerical flume in which wave direction of propagation and profile coincide, this is, of course, the best configuration to trace profiles. Nevertheless, in real scenarios is not so simple to define this direction. This aspect has been added in the new version of the manuscript (page 18)

7) The methodology is compared with numerical models and retrieves same estimations. The fact that inacessible high-resolution data can be overcame should be more highlighted. Again, I dont think Synolakis's formula is comparable here, since it uses a Solitary wave as initial condition, and there are analytical solutions that can handle with arbitrary shapes (Madsen & Schaffer,(2010) , Fuentes (2017) ). REPLY: The fact that inaccessible High-Resolution data can be overcome is now highlighted appropriately (Page26) and it has been included as well in conclusions section (page 41, line 16). AS explained in the reply to the comment 5, we have included the results of the application of Madsen formula, highlighting the fact, remarked by the reviewer 1, that Synolakis formula, although widely applied, was created for Solitary waves.

Specific comments: REPLY: Thanks a lot for this specific comments. We have added all of them to the new version of the manuscript. First line of intro: Add the 2010 Chile tsunami. Page 9, 5: It seems that "H" is unnecessary here. Also, it should be clarified that period relation is valid in the linear regime. Please add geographic axis to the map plots. Page 36, 5: Authors say "the results are accurate". Please, add a percentage based on the results.

Please also note the supplement to this comment:
https://www.nat-hazards-earth-syst-sci-discuss.net/nhess-2017-445/nhess-2017-445-AC1-supplement.pdf

---

## Author Comment (AC2) · 13 Feb 2018

Anonymous Referee #2 General comments: This paper presents a method for quickly assessing tsunami runup for different tsunami wave shapes and bathymetries. The method is built on a hybrid approach, where the Non-Linear Shallow water model COMCOT is used for the deep-water propagation and a RANS models is used for the near shore processes. The results from this hybrid model, then enters an interpolation model, which can be used to assess run-up. The approach is novel and innovative, and I especially like adding a fast interpolation model. I have however, a few major concerns. The actual implementation of the hybrid model is not well described and I think the coupling between the two models can pose big problems. Further, the hybrid model is not validated on its own in a controlled environment.

REPLY: We thank Reviewer 2 for their careful review and for the positive comments regarding the developed interpolation modeling. We have considered their modifications and suggestions, what has increased its overall significance. Changes included following Reviewer 2 comments are in yellow color in the attached revised version. We respond in detail to their comments below. We especially thank their comments towards explaining more deeply the hybrid model implementation. Initially, we included some more details regarding this specific part of the work, that concentrated our efforts, like a specific validation. However, in the end, we decided to skip some of this data to keep the focus of the paper on the run-up calculation. Following Reviewer 2 comments we have reconsidered this aspect, and we have added more information to make it easier all the process to follow. This incorporation provides a substantially improved approach to the hybrid model.

Major comments: 1) Parts of implementation and usage of the hybrid model is not well described. a. What are typical grid sizes in the RANS model? Are these sufficient to handle the processes, which NLSW models cannot handle? Like wave breaking.

REPLY: The design of the domain of the RANs model (IH2VOF) followed 2 criteria.

First, there is a limitation of the model that do not allow grid more than 5499 cells in X dimension (nx<5499), and the ratio between dimensions must be constant ($r=\Delta x/\Delta z$=constant). In this case, and due to the different scale necessities on each dimension, the applied ratio was r=5/1. Therefore, the maximum length covered with RANS model was Lx=nx*$\Delta$x.

And second, to control and avoid false wave breaking, the Z dimension of the RANS model grid must be discretized in a minimum number of cells, satisfying the expression:

$\Delta z=[(K*H\_COMCOT)/(10*0.05)]*0.05$

Where K is a safety margin of the model=1.08 and $\Delta z$ is defined in the range (0.05< $\Delta z$ <1).

To sum up, in this sense, the model itself limits the length Lx and the grid size $\Delta x$. First, $\Delta z$ is calculated with Hcomcot, then $\Delta x$ is obtained $\Delta x= \Delta z*r$, and finally, Lx=nx*$\Delta x$. This approach results in values of Lx, depending on the Tsunami wave height of the COMCOT model. For Hcomcot=0.5, then Lx=1400 m. For Hcomcot =4.5m, then Lx=12400m.

Following this process, the grid size is enough to handle the processes that the LSWE model cannot.

b. What are typical Lx lengths?

REPLY: Lx=nx*$\Delta x$, and the value of $\Delta x$ depends on Hcomcot. In the same way.as it has been described, the RANS model limits in terms of number of cells drive the generation of the grid, and also the value of Lx. ., that is calculated following two rules: "Maximize the area" means to use all the available cells to cover the maximum length. And, since the number of cells is limited, we did not want to lose cells onshore far away from the flooded area. Thus, we pre-calculate a rough value of the run-up using COMCOT and approach a first value of the run-up. Taking this into account, typical values of the RANS model length are from 500 to 25000 m.

c) How are the boundary conditions for the turbulence mode?

REPLY: In the case of turbulent flows, IH2VOF numerical model uses smooth wall, log-law distribution for the mean tangential velocity. This aspect has been included in page 4 of the new version of the manuscript.

d) It is unclear how xcut is determined. In the paper two criteria is given. One is to maximize the area of the IH2VOF domain and the other is to ensure that flooding does not exceed the inland end. Regarding the first criteria, letting IH2VOF cover

the entire numerical flume would achieve that, but that is clearly not what is being done. Regarding the latter, I cannot see how the end position of the IH2VOF domain influences the position of xcut.

REPLY: AS it has been described, the limit of the RANS model in terms of number of cells drives the generation of the grid. The position where the models are coupled, xcut, is given then by the value of Lx. As commented previously, Lx is set trying to maximize the cells that are effectively used in the simulation. This important aspect of the hybrid model was not included in the first submitted version of the manuscript, but following reviewer 2 comments we have added it. Regarding specifically the first criteria, since the flume is non-scaled, it was not possible to cover the whole domain with RANS model due to computational restrictions, i.e., we cannot calculate the generation-propagation and inundation areas without assuming other limitations of scale. Moreover, offshore generation and propagation is well solved by LSWE model, where non-linearities are not relevant in the calculation.

e) One of the advantages of the hybrid model is that the RANS model can handle processes that the simpler COMCOT model cannot. One of such processes is the wave splitting into an undular bore, which can happen when the wave travels long in shallow water and this has been witnessed in many real life tsunamis. To be able to capture this effect xcut needs to be positioned sufficiently off shore. How is this ensured?

REPLY: Due to the exposed characteristics of the model it is not possible to extend the RANS model grid seawards more than what this limitation allows. Nevertheless, to avoid loss of processes on coupling, several coupling tests were performed. These tests were conducted reproducing the flume on scale, so a target simulation was performed using only the IH2VOF model. Lately, simulations of coupling between IH2VOF + IH2VOF and COMCOT + IH2VOF were performed and compared to single IH2VOF simulations. In both cases, IH2VOF cases coupling methods reproduce evolution of single IH2VOF simulations adequately. In the attached figure one of the conducted

tests is given. In this figure, it can be observed that, when the last part of the flume is calculated by means of IH2VOF model, the comparison between the different combinations of models is accurate in terms of run-up.

AS a consequence, the part of the processes that are not covered by LSWE model is incorporated in the last part of the flume, where the RANS model works.

f) To avoid reflection, the numerical flume of COMCOT is altered, to properly access the incoming wave. I have a problem with this approach. In reality, especially in cases with steep slopes, there will be significant reflection from the beach which will and should affect the incoming wave. This effect cannot be captured with the current approach. Further it is also unclear what would happen when the reflected wave from the IH2VOF domain meets the hard boundary between the two models. Will this cause additional reflections in IH2VOF domain?

REPLY: The artifice that has been applied to avoid reflection effects on the input wave focus precisely on avoiding that the reflected wave affects the tsunami wave between borders incident one. This unaltered wave is used to force the IH2VOF domain, in which simulation reflection on the beach is of course observed and considered for the runup calculations. Figure 6 of the manuscript tries to explain how this artifice corrects the possible reflection effect. Fig 6a shows the reflected wave and Fig 6b shows how the artifice works and the reflective effect is almost imperceptible. This figure quote has been improved in order to make clear the artifice intention.

g) The calculations of L does not match Fig. 4. E.g. Li is given as $L_i = 1/50 \tan(\beta 0)$. This will result in a very low Li. Further $Loff$ is given as $L_f + x_2$, but according to Fig. 4 it should be $L_{off} = L_f + x_2 + x_1$. Finally, there is no need for $\delta X$ in the calculations of Lf as it is present both in the denominator and the numerator.

REPLY: Regarding the calculation of Li, the expression in the manuscript is incorrect, as describe by the reviewer. It should be: $L_i = 50/(\tan\beta\_0)$ Regarding Calculation of Loff, again, we appreciate reviewer correction. It should be: $L\_off = L\_f + x\_2 + x\_1$

Finally, regarding the Lf calculation, we used "ceiling brackets" which is the largest integer less than or equal to X, commonly used in mathematics and computer science. Then, using $\Delta x$ both in the denominator and numerator, allows us rounding values to the order of $\Delta x$.

h) How is the run-up height determined in the IH2VOF model? In a VOF computation,the interface can span across several cells.

REPLY: The numerical model calculates for the last flooded cell the ratio that is actually flooded and provides the run-up in accordance. Indeed, due to the fact that the RANS model does not calculate directly the free surface but it tracks the changes in cell density, there are mainly 2 ways to tackle this calculation: Assessing the iso-surface, determining the contour where VOF function is 0.5 Calculating the water contained, accumulated in a column of the grid. In order to avoid diffusion, the quantity of water is added. In this case, after several "trial and error" tests, the second method has been applied, although in the end the difference were not serious. None of the methods is perfect but both of them provided a good approach. This aspect is included in page 4 of the new version of the manuscript.

Responses to comments regarding models coupling have been included in the paper section regarding the characteristics of the numerical flume.

2) The first validation case is performed by comparing the interpolation model to the hybrid model. This is an important and satisfying validation case, but I am lacking validation of the actual hybrid model. How will the model perform using the approach outlined in the paper e.g. in cases with both breaking and non-breaking waves running up a constant slope.

REPLY: Both, COMCOT and IH2VOF models are models that have already been successfully validated in the past. The validation of the numerical flume and the coupling of the models was made by comparing its results with those conducted by Synolakis (1987) and Baldock (2009). The scenarios of these experiments were calculated using

COMCOT, IH2VOF, COMCOT+IH2VOF (the complete numerical flume) and the results were compared to the results of the physical experiments. The attached figure shows the run-up obtained in this comparison and how the results of both series of experiments fit adequately with the numerical flume results. This validation allowed us to continue with the database elaboration, and the database itself was then validated by comparing the results with both numerical models and field work data.

IH2VOF model is, in the hybrid model, the responsible of incorporating non-linearities and breaking effects. In this sense, apart from the validation as part of the numerical flume, it has also been validated and applied in many studies e.g. The shown figure, together with an explanation of the validation has been added to the manuscript, at the end of section 2 (pages 13).

3) The performance of the iterative solver is compared to the Synolakis formula. I do not believe this is a fair comparison, as it is created for the run-up of a solitary wave, which as highlighted by the author does represent a real geophysical tsunami event. A more relevant comparison could be to the analytical model proposed by Madsen and Schäeffer (2010), as also highlighted by the authors in the introduction. (The Synolakis formula require a proper reference).

REPLY: We appreciate this comment. It was also highlighted by Reviewer 1. We have added the results of applying Madsen and Shaffer (2010) to the table where the result of applying the run-up database interpolation are given. AS explained to Reviewer 1 we think that this complementary validation definitely improves the significance of the paper. In addition a proper reference to Synolakis formula has been included. We have decided to leave the Synolakis formula application results as well due to its common use in literature.

4) The periods are estimated as the time between the first two zero crossings for positive heights. Does this mean that the model cannot differentiate between tsunamis having only positive surface displacement and e.g. a leading depression?

REPLY: This criterion (period=time between 2 positive zeros), is the one that IHTRUST, the interpolation tool, and system, uses to automatically calculate the period of a time series. However, the system itself allows to manually edit the period: It shows the time series and the part of it that is going to be considered in the interpolation. If another period, like in the case of a leading depression is to be used, it can be manually corrected. However, several tests were performed, and we did not found serious differences on simulations led by crest or trough regarding the run-up results. Therefore, the tool was scripted following the explained criteria, although as explained it allows its modification.

5) With this approach of estimating period and wave height, I see a potential problem in the case where the leading wave is not the largest. Can you please elaborate on this?

REPLY: This is a real limitation that concerned us during the process of scripting the IHTRUST. It was the main reason why we added the "manual way" to include the height and period. This allows using an ad hoc input data on the interpolation process. Although obviously it cannot take into account several waves of the tsunami, it assures to use the proper part of the time series as input. We have reinforced the IHTRUST explanation on page 20

6) One of the main points off this work is to be able to quickly access tsunami runup without doing long complicated simulations. Therefore for this work to fulfill this, it would be beneficial is the TRD database was made available to engineers. Are there any plans regarding this? REPLY: So far, we are still making the most out of the tsunami run-up database. We are specifically working on a better definition of the influence of each parameter on the final value of the run-up. Once all the analyses are finished, we are planning to release the data, by itself or on a new issue.

Smaller comments: Thanks a lot for this smaller comments. We have included them in the new version of the manuscript. Page 1, line 8: It is stated that Run-up is accurately calculated by means of numerical models. This is a rather strong statement. I would

prefer it rewritten as: can be accurately calculated

Page 1, line 14. The models here, and several other places are referred to as schemes. They are however not schemes, but models. Please change the formulations. REPLY: We have changed it.

Fig. 5. It is stated that only the COMCOT model is used with the altered domain. If this is indeed the case, then please remove the IH2VOF domain from the figure, as it is causing confusion.

Fig. 6 units and legends are missing on the colorbars. Please add these.

Page 14, line 2. It is stated that d2 −d1 was always shorter than 2200 m and x1 shorter than 210 km. How does this correspond with table A1 where d2 −d1 is always larger than 2200 m? It should say larger.

Fig 8. Many of the axis are missing units. Please add these.

Fig 10. Please add missing units to the axis.

Page 19, line 6. It is stated that T corresponds to the time between the first two zero crossings for positive heights. However in Fig 11. It looks as if the second zero crossing has not occurred within the shaded area? The IHTRUST tool calculates the best fitting to this criteria. In this case the red shaded area approximates the period. The time series is built with discrete points and the system catches the closest one.

Page 24. It is described how low values of tan($\beta$2) gives lower run-up height due to friction. From Fig 13, it can however also be seen that the run-up heights reduce with large values of tan($\beta$2). Please elaborate on this.

It is unclear exactly what Fig 15 is describing. Please rewrite the description for clarity and add units to the axis. REPLY: We have modified this figure in order to make it clear that it is just a scheme of the main "regimes" that have been found. It is the figure 17 in the new version of the manuscript

Please also note the supplement to this comment:
https://www.nat-hazards-earth-syst-sci-discuss.net/nhess-2017-445/nhess-2017-445-AC2-supplement.pdf

―――――――――――――――――

[Figure]

**Fig. 1.**

[Figure]

[Figure]

[Figure]

Fig. 2.

**Supplement:**

[revised manuscript text omitted]

---

## Author Response (AR2)

Dear Dr. Didenkulova,

Please, find in this document, our response to reviewer 2 in which we have included detailed replies to their comments.
We have re-edited the manuscript, which is included at the end of this document. The changes are highlighted in yellow.

Íñigo Aniel-Quiroga
Corresponding autor

In the author reply, many of the details I requested are given, but a lot of them are not included in the revised manuscript, thus leaving a potential reader with the same questions.

The coupling between the two models and the setup of the IH2VOF should be dealt with in more detail in the revised manuscript. I still see some challenges in the coupling the two models. This, however, can be overcome by stating some of the possible limitations. Please see the detailed comments below.

REPLY: In the following paragraphs, the response to the reviewer detailed comments are given. As highlighted by the reviewer, a number of the comments were answered in the previous response, although not included in the manuscript. Now we have included all of them in the text and we have added the details requested in the previous revision response, and we have tried to efficiently explain both the IH2VOF and the coupling of the 2 models.

Detailed comments:

1)      I asked for typical grid sizes in the RANS model. This is given in the reply and I feel this information should be present in the paper.

The discussion detailed in the previous "authors response" has now been added at the end of 2.3.

a.      Further the authors state that the model do not allow more than 5499 cells in X. Why is that? Is that a choice of the authors?

The commercial version of the numerical model gives this recommendation. We accepted it after several tests in order to avoid long computational times when creating the database. (This aspect is also included in section 2.3).

b.      It is stated that the aspect ratio between x and z was 5. I completely understand the need for such a large aspect ratio, to limit computational time. However, large aspect ratio has, as shown by Jacobsen et al (2012), an impact on the position of the breaking point. Despite this, I still think that IH2VOF handle more accurately the physics of the tsunamis in this region than COMCOT, and thus it is justified. A comment should however be made that such large aspect ratios, might lead to slightly premature breaking.

We agree on this limitation, and it has been included also in the section 2.3.

c.      An equation is given for determining delta z. Where does this come from? The effect of it can clearly be seen, and to me it seem a reasonable way of automizing the process. Again, I feel that this information should be present in the revised manuscript.

Thanks a lot for this comment. The equation is a recommendation of the model and it is now explained in the manuscript, making the whole design process clearer.

2)      I asked for boundary conditions for IH2VOF model, and the authors replied that a log wall distribution was used. This seem reasonable, but, if the mesh is not graded near the bed, the $y+$ values must be extremely large potentially putting the value of the first grid point outside the log layer. Please discuss the possible impact of this. Further, please also provide

the wall functions for k and epsilon. Finally, another reference than Lara et al. (2006) should probably be used here, as the turbulence model is not described in this paper.

In addition to Lara et al (2006), we have added now other references that help to understand the IH2VOF turbulence model: Hsu et al (2002); Garcia et al (2004), Lin and Liu (1998, 1999)

In this references, specially in García et al(2004),  it is set that the model considers a log-law distribution of the mean tangential velocity in the turbulent boundary layer near the solid boundary, where the values of k (turbulent kinetic energy) and e (dissipation rate of turbulent kinetic energy) can be expressed as functions of the distance $y$ from the solid boundary and the mean tangential velocity outside the viscous sublayer. The grids must follow literature validations (Torres et al (2007, 2009), Lara et al (2011)) to set cell dimensions in order to avoid that the first grid point falls out of the log layer.

On the free surface, the zero gradient boundary conditions for both k and e are based on the assumption of no turbulence exchange between the water and air. The equations of the k-ε turbulence transport model are:

Free surface: no flux condition $\quad \frac{\partial k}{\partial n} = \frac{\partial \varepsilon}{\partial n} = 0$

Solid wall: log-law turbulent boundary layer for smooth surface $\quad \frac{\vec{u_n}}{u_*} = \frac{1}{k}\ln(\frac{Eu_* y}{\nu})$

Where n is; u is velocity ; y is distance from solid boundary  and $\nu$ is viscosity

These details regarding turbulence model in IH2VOF have been Included in the introduction of section 2 and in 2.3 in the new version of the manuscript.

3)      I asked how the x_cut positions was determined. The authors has given a clear and satisfying answer, but not included this explanation in the revised manuscript. Please do so. Further in the response the authors called the model a LSWE model. I thought it was a NLSW model. If it is a NLSW do not alter anything, but if it is a LSWE model, please justify why this sufficient for the simulations since tsunami close to the shore can definitely be nonlinear.

We have added the previous answer to the new version of the manuscript in 2.4. The model is NLSW. This has been corrected as well.

4)      I asked about the position of x_cut in relation to capturing the physics of the tsunami. A figure is added in the reply, but this figure the lines cannot clearly be distinguished and thus is hard to read. Further, I do not feel that this figure answers the question posed. One of the physical features a NLSW model cannot handle, but a VOF model can, is the undular bores. These might show further offshore than x_cut. I understand the practical limitation, but feel that a comment stating that there might be situations where the NLSW model is not properly handling the physics, before the VOF models takes over, is warranted.

We agree. One of the reasons of maximizing the RANS model domain is precisely to avoid those effects. A sentence regarding this limitation has been added to the manuscript in the section 2.4

5) I asked about reflection between the two models. I appreciate what the authors are trying to do, and can see that using the unaltered wave will limit reflection between the models. However, I think that this approach will give difficulties in certain cases. It the beach is steep, the tsunami wave will be reflected entirely, as the steep beach acts more or less like a vertical wall. In this case a standing wave will be present similar to that shown in Madsen and Fuhrman (2009) figure 9a. By using the unaltered waveform this behavior cannot be captured. This should be reflected upon in the revised manuscript.

We think that there is some confusion in this aspect and it is our intention to clarify it in the new version of the manuscript. The effects of the reflection are represented in the flume, both at COMCOT and IH2VOF domains.

The problem arises because in most cases tsunami wavelength is longer than IH2VOF domain, producing that the reflected wave on the beach reach the X-cut position before the incident tsunami wave completely cross this position, aliasing the signal to force the IH2VOF domain. Therefore, to eliminate this aliasing we designed the artifice described on the paper. This artifice is just applied to obtain the forcing wave to be used on the IH2VOF domain. Once the initial wave condition is obtained IH2VOF model is forced, reproducing correctly the reflection process observed on the coast.

The effects of the reflection are then taken into account in the run-up calculation. However, we have not analyzed the evolution of the wave and the interaction among several tsunami reflected waves, since we considered that it is out of the scope of the paper, but an effect like the considered standing wave generation could actually occur, and it would not be captured by the flume. We have added a sentence in the new version of the manuscript to explain this limitation.

6) I asked for more details regarding the calculations for figure 4. These have been provided, but some things are still unclear. Now it is stated that $L_i=50/\tan(\beta_0)$. How can this be true? In page 10 lines 17-19 it is stated that horizontal length of the domain comes from a separate simulation using COMCOT only.

The design of the flume was carried out by applying the expression $L_i=50/\tan(\beta_0)$. The formula´s objective is to set a limit of elevation of 50 m in order to define Li and $\beta_0$ from the cut with the profile. Then, in order to maximize the area where VOF model is applied, the run-up is pre-calculated with COMCOT and the VOF domain is moved seawards cover, with a safety factor to prevent that if run-up calculated with IH2VOF is longer that the one calculated with COMCOT, it does not reach the limit of the domain. As a result, the area where VOF model is effectively used (no-cells further than run-up limit) is maximum.

This explanation is included in the section 2.4 of the manuscript.

7) I asked for a validation of the Hybrid model. I fully appreciate that both models have been used with great success in the past, and I did not question the validity of the models. In Lara et al. (2006) however smaller aspect ratios were used compared to here. The additional figure indeed provides validation for that IH2VOF can handle run-up. It is however important that the mesh for these simulations were performed using the same rules as in the present paper. I.e. delta z= (K H)/10 *0.05) *0.05, and r=5/1. Is this the case? If not, I would argue that

the validation is made on a much finer mesh than the simulations in present paper. In this case, I do not feel that the model is validated satisfactory, and new validation simulations should be made.

The accompanying text to the figure is also unclear. In page 13 line 4 it is stated that the validation and coupling of the numerical models was made by comparing to experimental results. In the legend in the figure however, it is only the IH2VOF model, and thus no coupling between the models are present. Which of these are correct?

We appreciate this comment. We realized that the explanation could be a bit confusing.

As highlighted by the reviewer, Lara et al (2006) used smaller ratios than us. The reason for this is that in our case, tsunami problem requires some specific conditions for simulation due to the size of the domain and the wave, in particular the wavelength and period.

The validation process performed was complex:

First, to perform the validation, and due to the difficulties on using lab or real measurements of tsunami wave propagation we compared the results to the available experiments of Synolakis, Baldock, We verified that IH2VOF can reproduce Synolakis and Baldock experiments using the same rules as in the present paper. We have included this information in the paper accordingly.

In addition, we compared the run-up calculated with the numerical flume (COMCOT + IH2VOF) with the result of applying just IH2VOF in the whole geometry. We consider that IH2VOF model solves appropriately the whole domain, and thus the comparison with this performance at scale is an adequate validation. This comparison was depicted in a figure included in our previous "author´s response" and now, a modified and clearer version of it has been included in the new version of the manuscript.

[Figure]

We have modified a paragraph to explain these aspects of the validation process at the end of section 2.

8)      The authors have compared now both to Synolakis as well as Madsen and Schaëffer. I am however curious how the results from Synolakis as well as Madsen and Schäeffer were obtained, as these were created for only a single sloped region. Was an average slope calculated? Or only beta_1 or beta_0 used? Please describe how it was done, and why this is the best approach.

The application of these formulae to real geometry is, as the reviewer comments, at least complicated. One of the objectives of these methodology is precisely to overcome these difficulties. The validation of the methodology is made both with numerical models and recorded data. The comparison with Synolakis and Madsen&Schaëffer formulae is not to validate the methodology but to compare with existing alternative approaches.

There are, at first sight, several approaches to apply Synolakis and M&S formulae to our geometries. The direct use, as explained, is not easy, since there is not an *agreement* for their application. We find that the definition of the profile to use in the application of the formulae is complicated, mainly by the assumption of using a single slope to represent a more complex geometry. Considering our geometry, there are several options to apply the formula: using just beta_0, using just beta_1, using an averaged slope of beta_0 and beta_1, using an averaged beta from beta_0_1_2, etc. The use of a unique slope (beta_0 or beta_1) modifies strongly the profile. We studied several options and finally we observed that the approach that calculated the run-up more precisely (closer to the result of the numerical model) was using an averaged profile (from beta_0, beta _1, beta _2). We realize that this approach is not ideal, but it is the one that obtained the best results, and we chose it because we considered it the most coherent and the most restrictive in the comparison.

In the new version of the manuscript it is stated that the calculation with the formulae was made assuming an averaged slope.

9)	I asked about the determining of period and wave height. In the reply the authors state that no serious differences between trough led or crest led were experienced. What about difference between a single wave, and a leading depression n-wave for instance. With the N-wave height is the summation of the positive negative amplitude, whereas the wave height of the single wave is just the positive amplitude. I can see how a single wave with a similar positive amplitude as a leading depression N-wave, will run-up similar but I would imagine a single and an N-wave with the same wave height will run-up differently. Perhaps it would be better to called maximum positive amplitude in the revised manuscript rather than wave height.

We agree that the shape of the wave, its height and period, makes the final result to vary. However, as explained, the system itself allows to manually edit this values. In this sense, in order to be strict, in the definition of the height within IHTRUST, in section 4.1, the name of the automatically calculated height has been changed from wave height to maximum positive amplitude.

[revised manuscript text omitted]
$, what led to IH2VOF grid lengths between 500 m and 25000 m. And second, to control and avoid false wave breaking, the Z dimension of the IH2VOF model grid is discretized in a number of cells, satisfying the expression:

$$\Delta z = \left\lceil \frac{K \cdot H_{COMCOT}}{N_c \cdot 0.05} \right\rceil \cdot 0.05$$

15  Where $K$ is a safety margin of the model $K$ =1.08 and $\Delta z$ is defined in the range (0.05< $\Delta z$ <1). In this case, the wave height was discretized in $Nc = 10$ cells to avoid false breaking. The effect of the ceiling brackets is "rounding to the lowest integer". In addition, IH2VOF grids must follow literature validations (Torres et al (2007, 2009), Lara et al (2011)) to set cell dimensions in order to avoid that the first grid point falls out of the log layer.

20      **2.4  Numerical models coupling**

The coupling of the numerical models was focused on accurately locating the border position between the models, x$_{cut}$ (see Fig. 4). This location is optimized in the domain of the IH2VOF model for every tsunami scenario, since that area is the most computationally demanding. Two criteria were followed for this optimization: 1) maximize the area of the IH2VOF domain and 2) simultaneously ensure that the flooding does not exceed the inland end of the IH2VOF domain. In this sense, the number

25  of cells of the IH2VOF model drives the generation of the grid and the position where the models are coupled, $x_{cut}$, is given then by the value of $L_x$. In addition, since the flume is non-scaled, it was not possible to cover the whole domain with RANS model due to computational restrictions (i.e., the generation-propagation and inundation areas cannot be calculated without assuming other limitations of scale) but offshore generation and propagation is well solved by NLSWE model, where non-linearities are less relevant in the calculation. In addition, NLSW model do not calculate accurately some physical effects, like

30  undular bores, and due to this, IH2VOF domain is optimized in order to minimize this limitation.

[revised manuscript text omitted]

---

## Author Response (AR3)

Dear Dr. Didenkulova,

Please, find in this document our response to reviewer 2 comments.
We have re-edited the manuscript, which is included at the end of this document. The changes
are highlighted in yellow.

Íñigo Aniel-Quiroga
Corresponding autor

General comments:

The implemented changes are not as well written as the remainder of the manuscript.

REPLY: The changes have been reviewed, and also marked in yellow in this version of the manuscript.

Furthermore, only some of you equations are numbered. Please make sure to correctly number all the equations.

REPLY: Equations have now been properly numbered.

Beside this, I only have a few minor comments.

1) Page 11 line 11 please rephrase the sentence. E.g. "… Jacobsen et al., 2012) it limits the computational times relative to those achieved with smaller aspect ratios". Changed
2) Page 11 line 12: please add "the" in front of RANS. Added
3) Page 11 line 13: please replace what with which. Replaced
4) Page 11 line 15: Please provide units for Δz. (meters, now included)

5) Page 11 line 28-30: My earlier comment in relation to the NLSW models are not that they are bad at handing non-linearities, but rather that they cannot describe physical dispersion, which become important in the formation of undular bores. I would not necessarily say that the IH2VOF domain is optimized in order to minimize this limitation. How well the setup handles this situation will probably depend on the local steepness of the tsunami when the IH2VOF domain take over. Please slightly rephrase the sentence to take the above into account.

REPLY: We have rephrased the sentence to include reviewer´s comment. (Page 11, lines 28-31)

6) Regarding the wall functions for the turbulence model, I am happy to see it now in the manuscript. As you write in the reply, these kind of wall functions should be used for cells inside the logarithmic layer. Since you have not described any vertical near wall grading, I must assume that the first cell has the same height as the remainder of the domain. That can lead to very high y+ values potentially far outside the logarithmic layer. Please reflect upon this.

REPLY: We agree. We have included this limitation in the paper, highlighting the necessity of taking this into account when the IH2VOF model grid is constructed. (Pg11, line 18).

7) Page 4, line 28. Please define E. It is a constant, equal to 9.0 for smooth wall. It has been included in the manuscript.

8) Why was a smooth wall function used? I would imagine that the most situations would be hydraulically rough.

REPLY: Each local study case has its own conditions of "roughness", and so, an assumption was needed for the tsunami run-up database construction, and a smooth wall function was adopted as a simplification. Mentioned in page 4, line 29.

[revised manuscript text omitted]